# Self-amplified Amazon forest loss due to vegetation-atmosphere feedbacks

Delphine Clara Zemp[1,2,†], Carl-Friedrich Schleussner[2,3], Henrique M. J. Barbosa[4], Marina Hirota[5,6], Vincent Montade[7], Gilvan Sampaio[8], Arie Staal[9], Lan Wang-Erlandsson[10,11] & Anja Rammig[2,12]

Reduced rainfall increases the risk of forest dieback, while in return forest loss might intensify regional droughts. The consequences of this vegetation–atmosphere feedback for the stability of the Amazon forest are still unclear. Here we show that the risk of self-amplified Amazon forest loss increases nonlinearly with dry-season intensification. We apply a novel complex-network approach, in which Amazon forest patches are linked by observation-based atmospheric water fluxes. Our results suggest that the risk of self-amplified forest loss is reduced with increasing heterogeneity in the response of forest patches to reduced rainfall. Under dry-season Amazonian rainfall reductions, comparable to Last Glacial Maximum conditions, additional forest loss due to self-amplified effects occurs in 10–13% of the Amazon basin. Although our findings do not indicate that the projected rainfall changes for the end of the twenty-first century will lead to complete Amazon dieback, they suggest that frequent extreme drought events have the potential to destabilize large parts of the Amazon forest.

[1] Department of Geography, Humboldt Universität zu Berlin, Rudower Chaussee 16, 12489 Berlin, Germany. [2] Potsdam Institute for Climate Impact Research, P.O. Box 60 12 03, 14412 Potsdam, Germany. [3] Climate Analytics, Ritterstr. 3, 10969 Berlin, Germany. [4] Instituto de Física, Universidade de São Paulo, Rua do Matão 1371, 05508-090 São Paulo SP, Brazil. [5] Department of Physics, Federal University of Santa Catarina, Campus Universitário Reitor João David Ferreira Lima, 88040-900 Florianópolis SC, Brazil. [6] Department of Plant Biology, Institute of Biology, University of Campinas, Rua Monteiro Lobato, 255 13083-862 Campinas SP, Brazil. [7] Department of Palynology and Climate Dynamics, Albrecht-von-Haller-Institute for Plant Sciences, Georg-August-University, Untere Karspüle 2, 37073 Göttingen, Germany. [8] Center for Weather Forecasting and Climate Studies (CPTEC), National Institute for Space Research (INPE), Rodovia Pres. Dutra, km 39, 12.630-000 Cachoeira Paulista SP, Brazil. [9] Aquatic Ecology and Water Quality Management Group, Department of Environmental Sciences, Wageningen University, P.O. Box 47, 6700 AA, Wageningen, The Netherlands. [10] Department of Water Management, Delft University of Technology, P.O. Box 5048, 2600 GA Delft, The Netherlands. [11] Stockholm Resilience Centre, Stockholm University, Kräftriket, 10405 Stockholm, Sweden. [12] Department of Ecology and Ecosystem Management, TUM School of Life Sciences Weihenstephan, Technical University of Munich, Hans-Carl-von-Carlowitz-Platz 2, 85354 Freising, Germany. [†] Present address: Biodiversity, Macroecology & Conservation Biogeography, University of Goettingen, Büsgenweg 1, 37077 Göttingen, Germany. Correspondence and requests for materials should be addressed to D.C.Z. (email: dzemp@gwdg.de).

The Amazon forest has been listed as one of the tipping elements of the Earth system[1]. Large-scale vegetation shifts resulting from reduced rainfall probably occurred in glacial times[2,3] and might occur under twenty-first century climate change in combination with increasing deforestation, logging and fire[4–8]. It is an open question whether these stressors can trigger self-amplified forest loss in the Amazon basin[9,10]. Self-amplified forest loss may happen due to the strong coupling of vegetation and regional climate (Fig. 1). On the one hand, the vegetation state depends on the rainfall regime[6,11]. With decreasing precipitation, forest resilience (defined here as the ability of the forest to recover from perturbations[12]) decreases[12,13] and the forest might shift to an alternative low tree-cover (TC) state[11,14,15] as a result of perturbations such as fire and extreme drought events[8]. On the other hand, forest loss amplifies drought[16,17] by reducing dry-season evapotranspiration rates[18–20] and thereby weakening atmospheric moisture recycling, which is estimated to amount to 25–50% of total Amazonian rainfall[21–23]. As a consequence, a decrease in oceanic moisture inflow could trigger vegetation–atmosphere feedbacks and lead to self-amplified forest loss.

Intensification of the Amazonian hydrological cycle has been observed in the last decades, with the wet season getting wetter and the dry season getting drier in southern and eastern Amazonia[24–26]. This is partly explained by a reduction in oceanic moisture inflow caused by a sea-surface-temperature-induced northward displacement of the intertropical convergence zone[24,25]. Whether these anomalies will persist in the future is uncertain and climate model predictions in Amazonia vary from strong drying to modest wetting[27]. However, recent projections constrained with observations show a widespread drying during the extended dry season (June–November)[27]. Furthermore, while the spatial variability of precipitation during the Last Glacial Maximum (LGM, around 21,000 yr BP) was roughly similar to the present conditions, rainfall may have been lower over large parts of the Amazon basin[28–30] due to reduced dry-season oceanic moisture inflow induced by lower evaporation from the cooler sea surface[31].

Despite progress in recent years, the complex and nonlinear vegetation–rainfall interactions that may cause self-amplified Amazon forest loss are still poorly represented in process-based vegetation–climate models[32,33]. Here we provide a new perspective on the stability of the Amazon vegetation–rainfall system and the potential of self-amplified Amazon forest loss by applying a complex-network approach. Such an inter-disciplinary approach is powerful for analysing cascading effects and has been applied to study, for example, contagion in financial systems[34], the spread of innovations in society[35], catastrophic species extinctions[36], cascading failure in power grids[37] or the collapse of marine ecosystems[38]. Recently, complex networks representing statistical similarities in climatic fields were used to improve forecasts of Indian monsoon timing[39], extreme floods in the central Andes[40] and El Niño events[41]. Here, we use the reconstruction of moisture recycling networks[23] obtained from atmospheric moisture tracking[22,42] of synthesis climate data (see Methods). In these networks, nodes represent individual vegetation grid cells within the Amazon basin that are linked by monthly water fluxes from the source (evapotranspiration) to the sink (rainfall). Thus, rainfall in each node has an oceanic and a continental component[22]. The ability of such networks to represent real moisture recycling processes has been shown in a previous detailed analysis of networks' topology[23]. Combining moisture recycling networks[23] and an empirical indicator of forest resilience[11] in a unified cascade model[35] allows us to evaluate the strength and extent of vegetation–atmosphere feedbacks in a spatially explicit way while relying on observations.

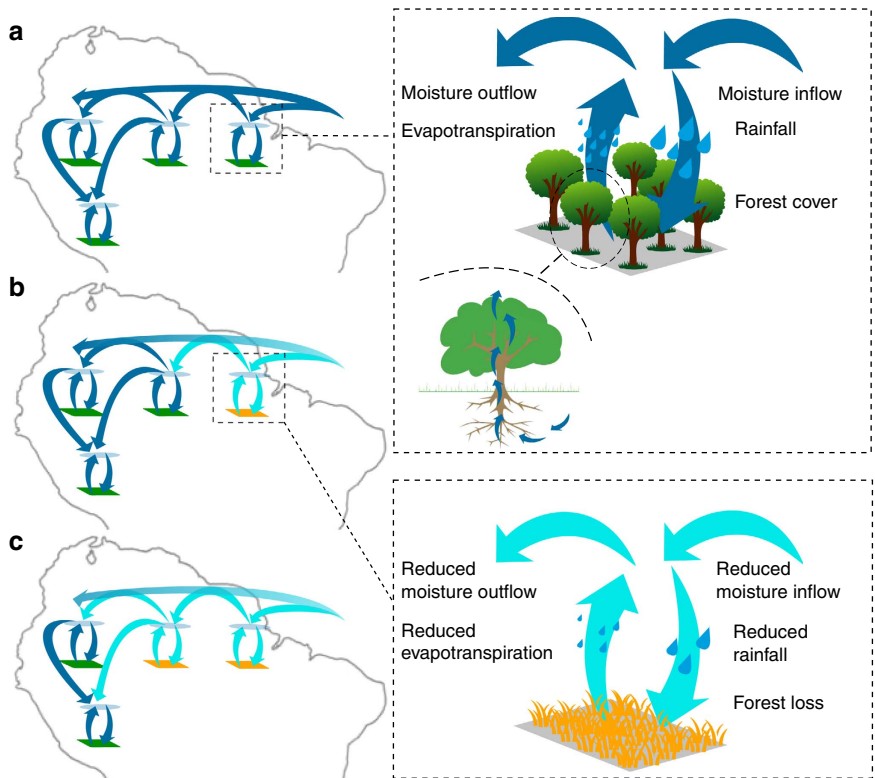

**Figure 1 | Schematic representation of cascading effects in the vegetation–rainfall system.** (**a**) Vegetation–atmosphere system in equilibrium. (**b**) Initial forest loss triggered by decreasing oceanic moisture inflow. This reduces local evapotranspiration and the resulting downwind moisture transport. (**c**) As a result, the rainfall regime is altered in another location, leading to further forest loss and reduced moisture transport.

## Results

**Forest shifts.** For each node, the rainfall regime is characterized by mean annual precipitation (MAP) and maximum cumulative water deficit (MCWD), a measure of the intensity of the dry season[43]. MAP and MCWD are well-suited climatic variables to explain the variability of vegetation distribution in the tropics[6]. Under a range of these variables, two TC states can be found (Supplementary Fig. 1): (1) an intermediate TC state ($5 \leq TC < 55\%$), comprising deciduous forest, shrubs and herbaceous and (2) a high TC state ($TC \geq 55\%$) corresponding to evergreen forest (hereafter simply called forest) (Fig. 2a). The probability of forest for different rainfall regimes is derived from satellite data (see Methods). This probability is used as an indicator of forest resilience[11] (Fig. 3a). Figure 2b shows that forest resilience decreases with reduced MCWD and MAP. In our interpretation, a forest with lower resilience is more likely to shift to a lower TC state in response to perturbations such as extreme drought and fire. In our modelling approach, forest nodes can shift stochastically based on thresholds in resilience (see Methods).

**Evapotranspiration and rainfall changes after forest loss.** Changes in evapotranspiration are evaluated using a statistical model based on multiple synthesis hydro-climate data (see Methods). The model accounts for the most important factors controlling monthly evapotranspiration in the Amazon basin as identified by flux tower measurements[19]: atmospheric demand (monthly potential evapotranspiration) and access of subsurface water during seasonal drought (carry-over factor). A complete Amazon deforestation experiment allows us to compare our estimations with existing studies. The spatio-temporal variability of evapotranspiration changes (Fig. 3b, see also Supplementary Figs 2 and 3) is in line with simulations from a recent mesoscale land-surface model[17] and measurements from flux towers[20]. Our estimates of mean annual evapotranspiration change for complete Amazon forest loss ($-108$ mm per year, see Supplementary Table 1) is at the lower end of estimates from multiple recent climate models ($-110$ to $-510$ mm per year, median $-219$ mm per year)[44]. Likewise, our estimates of MAP change over the Amazon basin ($-32$ mm per year) are much lower than the values from regional (70–475 mm per year)[44] and global (140–640 mm per year, mean 324 mm per year)[45] climate models. However, the upper-uncertainty 95th percentile bound of our estimate ($-274$ mm per year, $-13\%$) corresponds much closer to the mean rainfall changes simulated by multiple climate

models ($-324$ mm per year, $-12\%$)[45,46]. Based on this knowledge, our upper uncertainty bound can be considered as a realistic estimate. This result is consistent across all data sets used in this study (Supplementary Table 1).

**Self-amplified forest loss under dry-season intensification.** We simulate cascading effects in the vegetation–rainfall system (see Methods and Supplementary Fig. 4) triggered by gradual reduction of the contribution of oceanic moisture inflow to total precipitation during the extended dry season (June-November, Fig. 3c). Self-amplified forest loss increases nonlinearly with decreasing oceanic moisture inflow (gray shadings in Fig. 4a). This results from (1) a nonlinear decrease of forest resilience (Fig. 4b), (2) a stronger reduction of evapotranspiration after forest loss (Fig. 4c) and (3) an increased contribution of moisture recycling to total rainfall (Fig. 4b) under reduced dry-season oceanic moisture inflow. These findings are robust for different evapotranspiration input data sets and models (Supplementary Fig. 5), as well as cascade model settings (Supplementary Fig. 6).

While initial forest loss induced by reduced oceanic moisture inflow is sensitive to the underlying resilience thresholds, the additional forest loss attributed to vegetation–atmosphere feedbacks is more robust (Supplementary Fig. 6). Under a breakdown of oceanic moisture inflow during the extended dry season, this additional forest loss amounts to between 11–19% of the Amazon basin, depending on the thresholds. Among all the underlying processes represented, the largest uncertainties arise from the estimated changes in evapotranspiration (Fig. 4c). Considering these uncertainties, the additional forest loss attributed to vegetation–atmosphere feedbacks could amount up to 25–38% of the Amazon basin, depending on the thresholds (light gray shadings in Fig. 4a and Supplementary Fig. 6).

**LGM and twenty-first century.** Climate simulations of the LGM[31] indicate that oceanic moisture inflow during that time was reduced at the end of the dry season, resulting in a mean Amazonian rainfall decrease of 50% during the extended dry season (see Methods). Estimated change in vegetation cover (Fig. 5a) results from a large-scale shift of forests in the southern and eastern part of the Amazon basin (Fig. 5b). Oceanic moisture inflow reduction leads to initial forest shifts in the south-eastern part of the Amazon basin (light blue regions in Fig. 5c), triggering self-amplifying forest loss in regions located further south and west (red regions in Fig. 5c). Similar patterns are found using different resilience thresholds (Supplementary Figs 7 and 8).

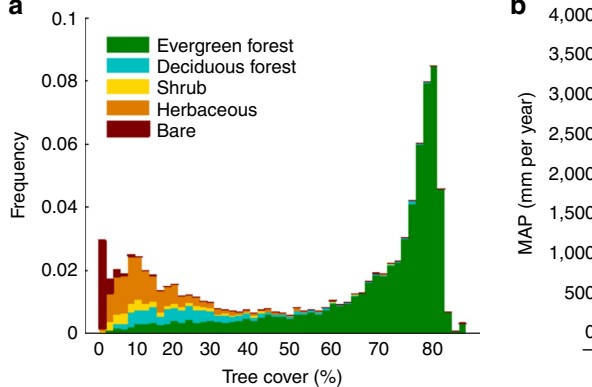

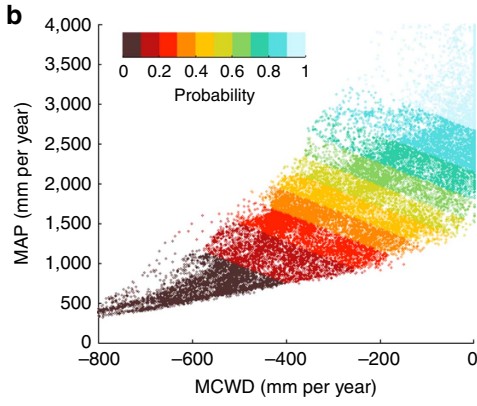

**Figure 2 | Probability of finding forest in tropical South America depending on rainfall regime.** (**a**) Frequency distribution of tree-cover (TC) data (MOD44B v5 for the period 2001–2010) and associated land-cover types (from GLC2000 classification). (**b**) Probability of finding forest (TC $\geq$ 55%) as a function of mean annual precipitation (MAP) and maximum cumulative water deficit (MCWD) calculated from a logistic regression model (equations 4 and 5) using monthly rainfall data (TRMM 3B42 for the period 2000–2012).

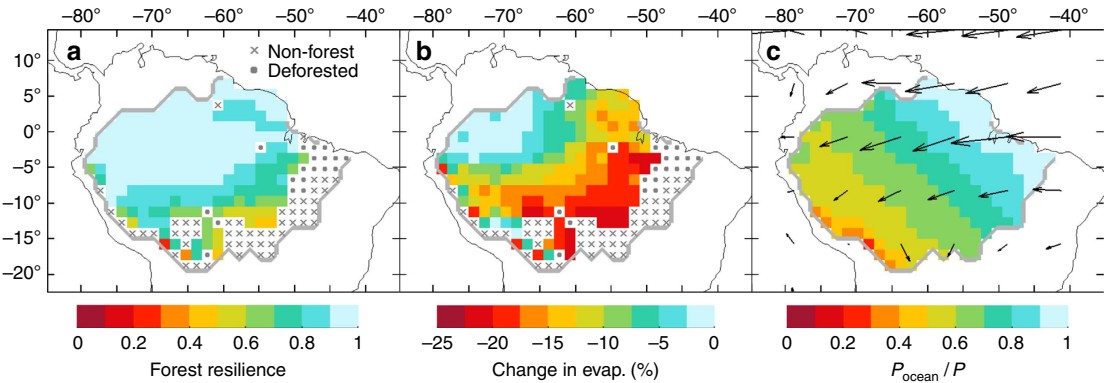

**Figure 3 | Vegetation–atmosphere coupling for current rainfall conditions in the Amazon basin. (a)** Amazon forest resilience calculated from a logistic regression model (equations 4 and 5) based on monthly rainfall data (CRU) for the period 1961–2012. **(b)** Simulated changes in evapotranspiration during the extended dry season (June-November) after complete forest loss calculated from a regression model (equations 15 and 16) fitted to synthesis hydro-climate data (1989–2005). The color scale has been truncated at 25%. **(c)** Fraction of total precipitation during the extended dry season (June–November) that last evaporated from the ocean, calculated from atmospheric moisture tracking[22,42] of synthesis hydro-climate data (1989–2005). Arrows represent vertically integrated moisture fluxes.

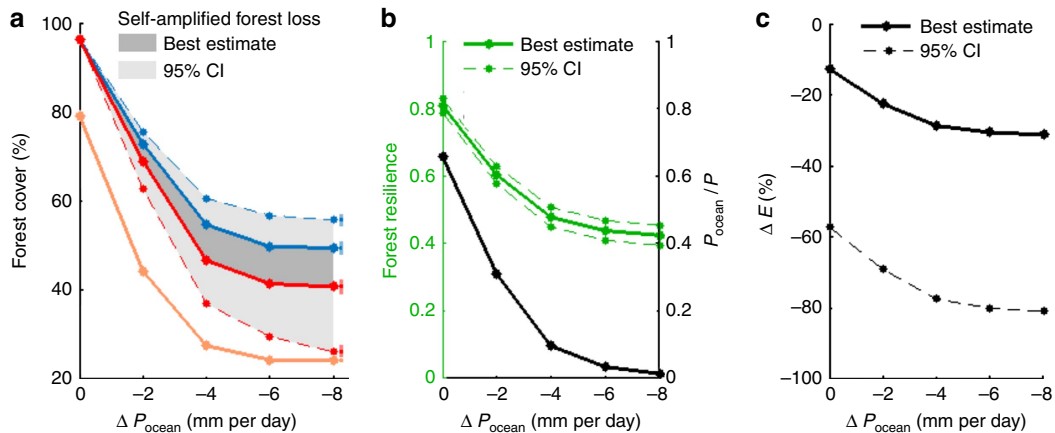

**Figure 4 | Self-amplified forest loss with dry-season oceanic moisture inflow reduction ($\Delta P_{ocean}$) and associated mechanisms. (a)** Fraction of the Amazon basin covered by forest (median of all 1,000 realizations of the cascade model as lines, 95% bounds shown by coloured bars on the right side of the plot). The difference between forest cover using the one-way coupling 'P→veg' (blue line) and the fully coupled system 'P↔veg' (red line) quantifies the additional forest loss due to self-amplifying effects (grey area). Results obtained considering the upper bound of the 95% confidence interval (CI) in estimated evapotranspiration change after forest loss are shown (dashed lines). For comparison, the maximum possible forest loss for a complete failure of moisture recycling is also shown (orange line). **(b)** Mean Amazon forest resilience (green line, left axis) and mean fraction of oceanic moisture that contributes to total rainfall (black line, right axis), calculated using the model version 'P→veg'. Whereas the 95% CI for the former is shown (dashed green lines), no error bars can be displayed for the latter but uncertainties associated with input data are shown in Supplementary Fig. 5a. **(c)** Mean evapotranspiration change ($\Delta E$) after complete Amazon forest loss, calculated using the model version 'veg→P'. The upper bound of the 95% CI of evapotranspiration reduction is shown (dashed black line).

Additional forest loss due to self-amplified effects for the LGM occurs in 10–13% of the Amazon basin and up to 18–23% when uncertainties in evapotranspiration estimates are considered (Supplementary Table 2).

A severe, stylized rainfall change scenario for the end of the twenty-first century following recent statistical projections[27], for which Amazonian rainfall reduction during the extended dry season is reduced by 40% induced by oceanic moisture inflow decrease, can lead to similar dynamics (Fig. 5c). Depending on the resilience thresholds, forest loss due to self-amplified effects occurs in 1–7% of the Amazon basin or up to 14% when uncertainties in evapotranspiration estimates are taken into account (Supplementary Table 2).

**Effect of heterogeneity.** Given the risk of self-amplified forest loss, understanding the specific properties and mechanisms that stabilize the system is of key interest. Previous modelling studies indicated the importance of spatial heterogeneity for the stability of ecosystems[47–49] and complex networks in general[35]. Here we assess the effect of heterogeneity in the response of forest nodes to changing climatic conditions on the stability of the Amazon vegetation–rainfall system. Theoretically, while a completely homogeneous forest would shift completely if it crosses a uniquely defined critical rainfall regime, heterogeneity in the forest response would result in forest nodes shifting at different critical rainfall regimes. Such heterogeneity may arise, for instance, from spatial variability in forest adaptability to drought, in land surface properties controlling water availability for trees and in disturbances to the forest. To evaluate the importance of such heterogeneity, we investigate the effect of the width of the bell-shaped resilience threshold distribution on cascade size (see Methods) under complete breakdown of dry-

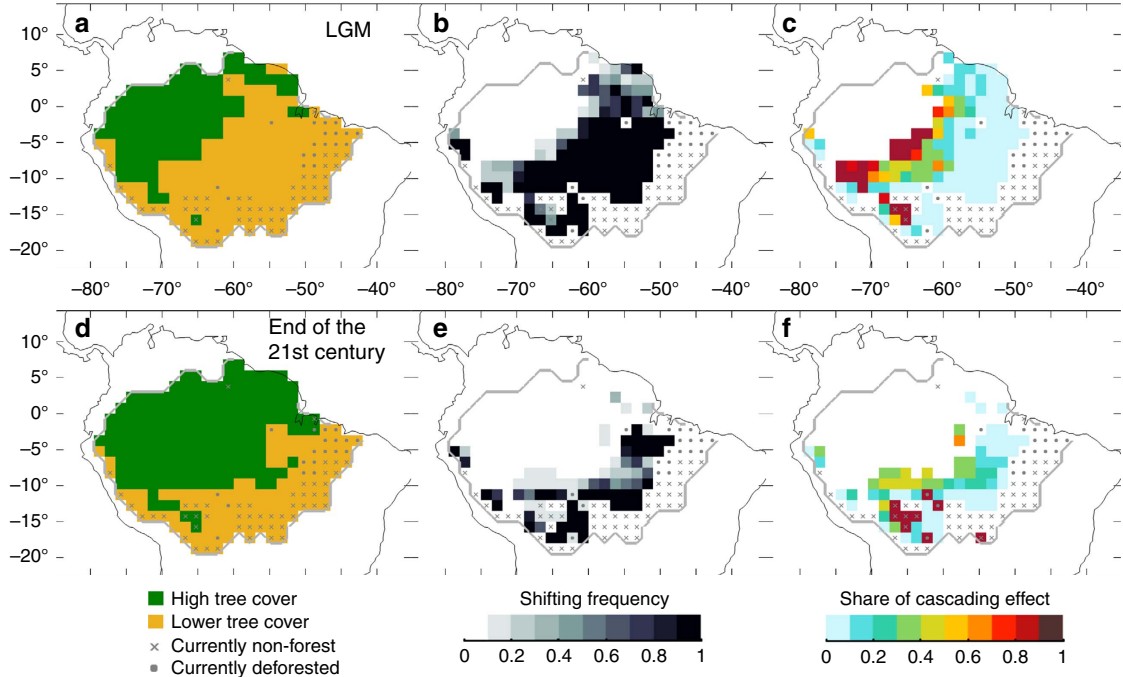

**Figure 5 | Self-amplified forest loss for the Last Glacial Maximum (LGM) and for the end of the twenty-first century.** (**a,d**) Most frequent vegetation cover for 1,000 realizations of the cascade model. (**b,e**) Shifting frequency of Amazon forest. (**c,f**) Share of cascading effects in causing forest shifts (see Methods). Results are shown (**a–c**) for the 'LGM' scenario and (**d–f**) for the 'end of twenty-first century' scenario (see Methods).

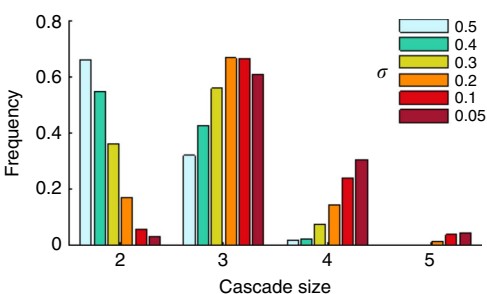

**Figure 6 | Effect of heterogeneity on the stability of the vegetation–rainfall system.** Frequency distribution of cascade sizes with increasing heterogeneity of the forest resilience thresholds based on 1,000 realizations of the cascade model. Note that with increasing heterogeneity ($\sigma$), the frequency of high-order cascades (cascade size = 4) decreases and the frequency of low-order cascades (cascade size = 2) increases.

season oceanic moisture inflow. Figure 6 shows that larger heterogeneity in forest resilience thresholds reduces the frequency of high-order cascades by more than 50%, regardless of the model settings (Supplementary Fig. 9). In other words, if individual forest nodes shift at different critical rainfall regimes, propagation of forest loss is usually stopped in the early stage of the cascade. Hence, variability in the Amazon forest's sensitivity to altered rainfall regime seems to alleviate the risk of long-term self-amplified forest loss.

**Sensitivity to evapotranspiration model and data**. Effects of forest loss and rainfall change on evapotranspiration are typically sensitive to the underlying model and input data[50,51]. To assess the sensitivity of our results, we performed additional analyses using a generalized linear model (equation (17)) and for different input evapotranspiration data (satellite observations and/or

measurements, simulations from land-surface models, output from atmospheric reanalyses and the merged synthesis of all these three categories[52]). We find that these choices do not affect our results. Firstly, the spatio-temporal variability of simulated evapotranspiration changes after forest loss is consistent for all input data sets and for the two statistical models considered (Supplementary Figs 2 and 3). Secondly, similar dynamics are found regarding the effect of decreasing oceanic moisture inflow on simulated changes of evapotranspiration after forest loss (Supplementary Fig. 5c,d). This gives us confidence that our main findings are robust with respect to different choices of evapotranspiration model and input data. However, the absolute changes might vary depending on the evapotranspiration data. The results shown in the main part of the manuscript are therefore based on the merged synthesis evapotranspiration data set.

## Discussion

Our results highlight the key role of regional dynamic vegetation–atmosphere interactions in the Amazon basin, which are not considered in most previous modelling studies assessing the likelihood of Amazon forest dieback for the future[5,6,53,54] or for the LGM[3,55]. As it does not resolve the underlying processes of forest dieback, our method is not suited to provide information on the 'real-world' time scale of self-amplified forest loss. Our study should be seen as a sensitivity analysis rather than a projection of the system dynamics, as it omits some key feedbacks important for regulating forest cover. In particular, wind fields are considered to be static, which is a possible drawback of our approach as forest loss is expected to alter atmospheric circulation patterns. However, this effect might be relatively weak over the Amazon compared to changes in the amount of water transported[17]. We do not account for the effects of changing temperature and atmospheric $CO_2$ concentrations on the hydrological cycle and on forest resilience, the latter remaining highly uncertain[6,53,54].

Our results for the LGM show a self-amplified expansion of the intermediate TC state (encompassing savannas and/or deciduous forests) at the expense of the high TC state (evergreen forests). Although we do not account for all processes of vegetation dynamics, this finding is in line with recent observations of Amazonian speleothem records[29,30] indicating a reduced moisture recycling during this period[29,30]. This gives us confidence that vegetation–atmosphere interactions, as represented in our modelling approach, probably play a major role in explaining Amazonian precipitation and vegetation changes. Furthermore, our results suggest that self-amplified forest loss triggered by oceanic moisture inflow reductions similar to LGM conditions did not affect the western Amazonia. A more resilient forest area related to more climatically stable conditions in this region, which is also suggested by paleoenvironmental records[3,29,30,56], might have been crucial in the biogeographic history of one of the world's most species-rich terrestrial ecosystems.

For the end of the century, in agreement with previous studies[53,54,57], we find that projected rainfall reduction does not lead to complete dieback of the Amazon forest. However, if extreme drought events become more frequent or intense in the future, as suggested by some studies based on current trends[25] and climate projections[51,58], these may push the system towards large-scale self-amplified forest loss in a step-wise process without similarly drastic changes in the long-term mean rainfall regime. Indeed, the main mechanisms responsible for large-scale self-amplifying forest loss identified in our study might occur during such extreme drought events. Firstly, we found that the reduction of evapotranspiration after forest loss is more pronounced with increasing water deficit, consistent with measurements from flux towers[20]. This is due to the fact that, compared to forest, vegetation states with lower TC are less able to access subsurface water and thereby to maintain high evapotranspiration rates during drought[18–20]. Secondly, we find that the contribution of moisture recycling to total rainfall increases with reduced oceanic moisture inflow, consistent with paleoprecipitation records[29]. These two effects have probably already occurred during recent drought years[17]. Further efforts are needed to assess the effect of inter-annual rainfall variability on the stability of the Amazon vegetation–rainfall system and potential time lags in the response of the coupled system.

By fully coupling vegetation and rainfall, our study goes beyond previous efforts combining moisture-tracking algorithms with statistical[16] or mechanistic[17] climate models. Existing coupled models still poorly represent nonlinear regional interactions between biosphere and atmosphere, as shown by large discrepancies in predictions depending on model structure and settings[9,59]. Misrepresented key processes include (1) access of subsurface water by tropical trees, leading to an underestimation of the reduction of evapotranspiration after forest loss (by around 1 mm per day)[55,60], (2) moisture recycling, leading to biases in moisture fluxes[61] and (3) forest responses to drought[5–7]. In our statistical model, all abovementioned processes are included by applying vegetation and hydro-climate data, which are either observation-based or merged from multiple data sets. Our findings are robust against choices of evapotranspiration model and data, to which insights about the hydrological cycle are usually sensitive[50,51].

This work adds to previous studies that suggested the importance of heterogeneity for ecosystems' stability[32,33,47,62,63] and the Amazon forests in particular[48,49]. However, none of these studies considered spatial interactions among vegetation patches[32,33,48,49,62] or interactions were represented non-realistically[47,63]. Our results illustrate the importance of maintaining the structural and functional diversity of Amazon forests to reduce the risk of long-term self-amplified forest loss. This should be incorporated into future conservation management strategies and calls for cross-regional and -national approaches.

## Methods

**Hydro-climate data.** The input data for the construction of moisture recycling networks and calibration of the evapotranspiration model cover the time period 1989–2005 for tropical South America (between 14.25° North and 23.5° South). The precipitation data are an average of four observation-based data sets: Climate Research Unit (CRU), the Global Precipitation Climatology Centre (GPCC), the Global Precipitation Climatology Project (GPCP) and the unified Climate Prediction Center (CPC) from the National Oceanic and Atmospheric Administration. A description of the precipitation data sets is provided in Appendix A in ref. 52. The potential evapotranspiration data are based on the Penman–Monteith equation which is forced by temperature, humidity and wind speed from reanalysis data corrected to remove biases at various time scales by merging with observation-based data[50,64]. The evapotranspiration data sets are taken from LandFlux-EVAL and are merged synthesis products from different categories: derived from satellite observations and/or measurements ('diagnostic'), simulations from land-surface models and output from atmospheric reanalyses. In the preparation of the merged LandFlux-EVAL data, the long-term evapotranspiration by the energy balance was constrained (latent heat flux cannot exceed net surface radiation) but not by the available water (mean annual evapotranspiration can exceed precipitation). Evapotranspiration, precipitation and potential evapotranspiration data sets are available at monthly time scale and for a 1° longitude and latitude grid. Consistencies between these data sets are shown by Budyko curves (Supplementary Fig. 10). Wind fields and specific humidity data are taken from the six-hourly ERA-Interim reanalysis product[65].

**Data for forest resilience.** Calculation of forest resilience is based on TC data from the Moderate Resolution Imaging Spectroradiometer (MODIS) Vegetation Continuous Fields MOD44B v5 (ref. 66) averaged for the period 2001–2010. The land-cover map is taken from the Global Land Cover 2000 (GLC 2000) Database[67]. To account only for natural distribution of the vegetation, human-modified landscapes and water bodies (GLC2000 classes 16–18 and 20–23) have been excluded from the analysis. TC and land-cover data are available at 1 km resolution of longitude and latitude, and have been sampled at the centroid of each precipitation grid cell.

For visualization of current forest resilience in the MAP-MCWD space (Fig. 2), precipitation data are derived from the Tropical Rainfall Measuring Mission (TRMM) 3B-42 v7 (ref. 68) for the years 2000–2012. These data have the advantage of their high spatial resolution (0.25° grid) and their ability to represent temporal and spatial variability over tropical South America[69–71]. For calibration of the logistic regression model to simulate self-amplified forest loss, precipitation data are derived from CRU, which have lower spatial resolution (0.5° grid) but longer duration (1961–2012).

**Vegetation–cover data for the evapotranspiration model.** Vegetation cover for the evapotranspiration model calibration is based on MOD44B v5 (ref. 66) TC data for the year 2001. To fit with the grid of the LandFlux-EVAL product, TC data have been upscaled to 1° resolution using the most frequent value. The Andes mountains were excluded from the analysis as evapotranspiration in this region is mainly determined by temperature rather than by rainfall. Artificial landscapes were not excluded. To exclude the Andes from the analysis, we use a natural-vegetation-cover map at 1° resolution that is based on a consensus of two global natural-vegetation-cover maps widely used in climate studies, as well as several regional maps from different sources[72]. The classes that represent the Andes region and that were excluded from the analysis are 'desert', 'semi-desert', 'tundra' and 'grasslands'.

**Vegetation-cover data for initial conditions in the cascade model.** The initial vegetation cover for the cascade model is based on two different data sets, depending on whether grid cells are within the Amazon basin (area of interest) or outside the Amazon basin (boundary conditions). Within the Amazon basin, initial vegetation cover is derived from satellite monitoring of the vegetation[73] for the year 2003. This data set distinguishes between forest (here set as 'high TC state'), non-forest (here set as 'intermediate TC state') and deforested area (here set as 'treeless state'). The initial vegetation cover outside the Amazon basin is based on MOD44B v5 (ref. 66) TC data for the year 2001. To fit with the grid of moisture recycling networks, both vegetation data sets were upscaled to 1.5° resolution by selecting the most frequent value found in the original data.

**Vegetation classification from TC data.** Following previous studies[11,14], TC data were used to classify treeless ($5\% < $ TC), intermediate TC state ($5\% \leq $ TC $< 55\%$) and high TC state (TC $\geq 55\%$) states. As the spatial distribution of the two former states might differ depending on the spatial resolution considered (Supplementary Fig. 11), we do not distinguish between these two states in step 4 of our cascade

model. Rather, shifts occur between high TC and lower TC states (TC < 55%). To estimate the effect of the shifts on evapotranspiration, we use parameters estimated for the treeless model (Supplementary Table 4).

**Moisture recycling networks.** Following a previous study[23], moisture recycling networks were built with the atmospheric moisture-tracking model WAM-2layers V2.3.01 (refs 22,42). As requested by WAM-2layers settings, all data have been spatially interpolated to a 1.5° grid using the nearest-neighbour algorithm. The temporal resolution of WAM-2layers is 3 h, to which monthly evapotranspiration and precipitation data have been downscaled using the temporal dynamics of ERA-Interim products. The output of WAM-2layers is averaged to monthly moisture transport between grid cells. Hence, in the networks, for each pair of grid cells (or nodes) $j$ and $k$ and for a given month, the weight of the arrow originating from $j$ and pointing towards $k$ ($m_{j,k}$) is the monthly amount of water that comes from evapotranspiration in $j$ and falls as rain over $k$.

For each grid cell $k$ and for each month, the sum of all incoming arrows ($\sum_j (m_{j,k})$) is the amount of precipitation over $k$ that originates from the continent. Hence, $1 - \sum_j (m_{j,k})/P_k$, with $P_k$ being the total monthly precipitation in $k$, is the fraction of total precipitation that last evaporated from the ocean (Fig. 3a). Note that it corresponds to $1 - \rho$, with $\rho$ being the continental precipitation recycling ratio[22].

**(Maximum) climatological water deficit.** The MCWD is the most negative value of the climatological water deficit ($C$). For each grid cell $k$ and for each month $t$ (ref. 6):

$$C_{k,t} = C_{k,t-1} + P_{k,t} - E_{fix}, \tag{1}$$

$$\max(C_{k,t}) = 0, \tag{2}$$

$$C_{k,0} = C_{k,12}, \tag{3}$$

with $P_{k,t}$ being the precipitation in month $t$ and grid cell $k$. The MCWD is an indicator of 'meteorologically induced' water stress[43] and therefore evapotranspiration is fixed ($E_{fix} = 3.3$ mm per day). $E_{fix}$ is an approximation of evapotranspiration rate under favourable climatic conditions, which we found to be reached both in high and lower TC regions.

**Forest resilience.** We calculate the resilience of the vegetation based on ref. 11, but including MCWD in the calculation as well instead of MAP only. MCWD was calculated on monthly rainfall data averaged for the entire period (rather than for each year individually) following a previous study[6].

The binomial distribution of forest (TC ≥ 55%) was fitted to a logistic regression model using the Matlab function 'glmfit':

$$f(\mathbf{z}) = \frac{1}{1 + \exp(-\mathbf{z})} \tag{4}$$

$$\mathbf{z} = \beta_0 + \beta_1 \mathbf{x} + \beta_3 \mathbf{y} \tag{5}$$

with $\mathbf{x}$ corresponding to MAP and $\mathbf{y}$ to MCWD. For statistical stability, data points with extreme hydro-climatic values ($\mathbf{y} = 0$ and $\mathbf{y} < 800$ mm) were excluded from the analysis. Estimates of the parameters are shown in Supplementary Table 5.

**Potential landscapes.** We estimated under which conditions high and intermediate TC are alternative stable states by performing potential analysis[74] on the TC data set following an existing method[11]. We computed stability landscapes of TC by determining how the probability density of TC changes with MAP (0 to 2,500 mm per year in steps of 25 mm per year) and MCWD (−800 to 0 mm per year in steps of 80 mm per year). At each step of the climatic variable (MAP or MCWD), TC values were weighted by applying a Gaussian kernel on the climatic variable with a standard deviation of 5% of the total range considered. Subsequently, the probability density of TC was estimated using the Kernel smoothing function in Matlab (ksdensity) with a bandwidth according to Silverman's rule of thumb. Maxima (minima) in these probability densities are considered to be stable (unstable) states, whereby local fluctuations in the densities were filtered out[11]. Forest is bistable under climatic conditions below the highest values of MAP and MCWD at which a stable intermediate TC state is inferred (respectively 2,000 and −88 mm per year) (Supplementary Fig. 1).

**Simple evapotranspiration model on a monthly time scale.** To estimate changes in local evapotranspiration with changing rainfall or vegetation state, we use a nonlinear regression adapted from a simple evapotranspiration model on a monthly scale, hereafter called 'Gerrits' model'[75]. Evapotranspiration ($\mathbf{E}$) includes evaporation of intercepted water by the surfaces (canopy, understory, forest floor and the top layer of the soil) ($\mathbf{E}_i$) and transpiration by the vegetation ($\mathbf{E}_t$), while evaporation from deeper soil and open water are neglected. In Gerrits' model, monthly transpiration is modelled as a simple threshold process that is a function

of monthly precipitation $\mathbf{P}$:

$$\mathbf{E}_t = \min(A + B_t(\mathbf{P} - \mathbf{E}_i), D_t) \tag{6}$$

where $A$ is a carry-over factor that represents the transpiration rate at $\mathbf{P} = 0$ and relates to the access of the vegetation to subsurface water during seasonal drought. It depends on vegetation-rooting depth and soil moisture. $B_t$ is the slope between effective rainfall (rainfall minus interception evaporation) and transpiration. This slope can be estimated as $B_t = 1 - \omega - \omega(\exp(-\omega))$ with $\omega = S_b/D_t$ where $S_b$ is the soil moisture below which transpiration is constrained. $D_t$ is the monthly potential transpiration (that is, the atmospheric demand for evaporation once the interception process has first absorbed its part of the available energy). In Gerrits' model, monthly evaporation from interception is modelled as:

$$\mathbf{E}_i = \mathbf{P}(1 - \exp(-B_i/\mathbf{P})) \tag{7}$$

where $B_i$ is the potential amount of monthly interception (in terms of storage capacity). The authors[75] also provide numerical derivation of the processes that takes into account the distribution of the expected number of rain days per month. For simplicity, we only use the analytical derivation.

Here, we assume that limits of $\mathbf{E}_i$ and $\mathbf{E}_t$ can be combined to derive the overall limiting factors of $\mathbf{E}$. Because there is no interception with no rainfall, $\mathbf{E}$ is limited at $\mathbf{P} = 0$ by $A$. In addition, the upper limit of $\mathbf{E}$ is the potential evapotranspiration $\mathbf{E}_p$. We can thus model $\mathbf{E}$ as:

$$\mathbf{E} = \mathbf{E}_i + \mathbf{E}_t \tag{8}$$

$$\mathbf{E} = \mathbf{E}_i + \min(A + B_t(\mathbf{P} - \mathbf{E}_i), D_t) \tag{9}$$

$$\mathbf{E} = \min(A + \mathbf{E}_i + B_t(\mathbf{P} - \mathbf{E}_i), \mathbf{E}_p) \tag{10}$$

$$\mathbf{E} = \min(A + \mathbf{P}(1 - \exp(-B_i/\mathbf{P})) + B_t(\mathbf{P} - \mathbf{P}(1 - \exp(-B_i/\mathbf{P}))), \mathbf{E}_p) \tag{11}$$

$$\mathbf{E} = \min(A + \mathbf{P}(1 - \exp(-B_i/\mathbf{P}) + B_t(1 - (1 - \exp(-B_i/\mathbf{P})))), \mathbf{E}_p) \tag{12}$$

$$\mathbf{E} = \min(A + \mathbf{P}(1 - \exp(-B_i/\mathbf{P}) + B_t(\exp(-B_i/\mathbf{P}))), \mathbf{E}_p) \tag{13}$$

$$\mathbf{E} = \min(A + \mathbf{P}(1 - \exp(-B_i/\mathbf{P})(1 - B_t)), \mathbf{E}_p). \tag{14}$$

To account for depletion of subsurface water, here we consider that $A$ is a variable that is modelled as a linear function of the climatological water deficit ($C$, see equations 1–3). Here, $C$ is estimated individually for each year (rather than on the average for the entire period). We call $p_1 = 1/(1 - B_t)$ and $p_2 = B_i$. Our evapotranspiration model becomes:

$$\mathbf{E} = \min\left(\mathbf{A} + \frac{\mathbf{P}}{p_1}\left(1 - \exp\left(-\frac{p_2}{\mathbf{P}}\right)\right), \mathbf{E}_p\right) \tag{15}$$

$$\mathbf{A} = -p_3\mathbf{C} + p_4 \tag{16}$$

with $p_1$, $p_2$, $p_3$, $p_4$ parameters. While these parameters depend typically on land-surface properties (soil and vegetation) and daily rainfall characteristics[75], for simplicity we consider here only the vegetation control. These parameters are estimated for each vegetation state (high TC, intermediate TC, treeless) using iterative least-squares estimation based on the function 'nlinfit' in Matlab with initial values set to 1.

The estimated parameters and the corresponding standard errors for each category of evapotranspiration input data set are shown in Supplementary Table 4. To account for uncertainties associated with the parameters, we perform simulations for which evapotranspiration rates in high and low TC states nodes are replaced by the 95% upper bound and 5% lower bound of the estimates, respectively.

**Generalized linear model.** To assess the sensitivity of our results to the choice of evapotranspiration model, we performed additional analyses using a generalized linear model:

$$\mathbf{E} = p_5\mathbf{P} + p_6\mathbf{P}^2 + p_7\mathbf{C} + p_8\mathbf{C} + p_9, \tag{17}$$

in which $p_5$, $p_6$, $p_7$, $p_8$, $p_9$ are parameters. Using a generalized linear mixed-effects model would be more appropriate due to dependency of the observations (spatio-temporal data). However, the use of a generalized linear model already reproduces sufficiently well the results based on our evapotranspiration model (equations 15 and 16).

**Cascade model.** Our approach has been largely inspired by the 'Watts model'[35], which simulates cascades that are found in social systems, such as the spread of innovations. Typically, individuals must decide between two alternative actions and their decisions depend explicitly on the actions of other individuals with whom they interact. The Watts model is based on a random network, in which each node might shift between alternative states depending on the states of their neighbours according to a simple threshold rule. Here, our model is based on moisture recycling networks and the threshold rule is related to local forest resilience that depends on the total incoming moisture. Initially, each node is assigned a

resilience threshold drawn from a normal distribution with mean $\phi$ and standard deviation $\sigma$.

A simulation run comprises the following steps (Supplementary Fig. 4). First, moisture of oceanic and continental origin propagates through the network using moisture recycling networks and the simple evapotranspiration model (equation 16). This occurs on a monthly time scale, where the strongest effects of varying vegetation states on evapotranspiration can be observed, as shown by flux tower measurements[18–20]. In this step, it is assumed that with the moisture recycling link between two nodes, $m_{j,k}$ (from $j$ to $k$) changes linearly with total evapotranspiration in the source node (in $j$). This implies that atmospheric circulation following vegetation shifts are not considered and that rainfall is assumed to be linearly correlated to atmospheric moisture. Second, the rainfall regime characterized by the MAP and the MCWD are calculated for each node. Third, the forest resilience is calculated for each node (equation 5). Fourth, a critical transition occurs in all nodes for which the forest resilience crosses the individual threshold, without any time lag. Fifth, local evapotranspiration is updated in nodes where shifts occur. The model runs with nodes updating their status until equilibrium in vegetation cover is reached.

The resilience thresholds are fixed for the duration of the simulation and potential spatial correlations are not considered. Results are shown for 1,000 random realizations of the initial condition. In the main text, it is assumed that $\phi = 0.5$, which best reproduces vegetation cover under current rainfall conditions (Supplementary Fig. 12), and $\sigma = 0.05$, which limits the occurrence of shifts (Supplementary Fig. 13) to the zone where high TC and intermediate TC states can both exist (Supplementary Fig. 1). Results for other plausible values of $\phi$, ranging from 0.4 to 0.6, are shown in Supplementary Figs 6–8.

We use different model settings: a fully coupled vegetation–rainfall system ('P↔veg'), a one-way coupled system in which changes in vegetation states do not affect precipitation ('P→veg') and a one-way coupled system in which changes in precipitation do not affect vegetation states ('veg→P') (Supplementary Fig. 4). Several metrics are of interest: first, the difference between forest loss in P↔veg and P→veg, which quantifies the additional forest loss due to self-amplified effects; second, the relative difference between the shifting frequency in P↔veg and P→veg, which quantifies the share of cascading effect in forest loss ('share of cascading effect'); third, the number of model iterations until equilibrium ('cascade size') in the fully coupled version (P↔veg). The cascade size can be interpreted as a time span of the cascade, but only on a relative scale (for comparison between high- and low-order cascades). We note that only nodes in which the shifting frequency exceeds 3% are accounted for.

**Experiments of dry-season oceanic moisture inflow reduction.** We present two different set-ups to study the effects of decreasing monthly oceanic moisture inflow during the extended dry season (June–November) with the period 1989–2005 as baseline. In a first setup, oceanic moisture inflow is homogeneously decreased by 2 mm per day increments until this inflow ceases completely in most of the Amazon basin (up to 8 mm per day reduction for each month of the extended dry season).

In a second setup, two stylized scenarios (end of twenty-first century and LGM) are generated based on long-term precipitation change estimations for the extended dry season in the Amazon basin drawn from previous studies[27,31]. The first scenario (end of twenty-first century) follows the upper bound of projected precipitation reduction for the end of the twenty-first century (2060–2099) based on a combination of observation-based data and the ensemble mean of CMIP5 climate models[27]. In this scenario, monthly oceanic moisture inflow is reduced homogeneously by [0.8, 1.0, 1.1, 1.3, 1.0, 0.8] mm per day (in June–November). If the resulting amount of oceanic moisture inflow becomes negative, it is set to zero. Averaged over the Amazon basin during the extended dry season, it corresponds to an oceanic moisture inflow reduction of 57% and a total rainfall reduction of 38% (calculated from step 1 of the cascade model).

The second scenario (LGM) follows a reconstruction of Amazonian rainfall during the LGM from a regional climate model, in which the effect of vegetation on climate has been turned off (Fig. 13b in Cook et al.[31]). Here, monthly oceanic moisture inflow is reduced by [0, 1, 2, 6, 10, 6] mm per day (June–November). Averaged over the Amazon basin during the extended dry season, it corresponds to an oceanic moisture inflow reduction of 76% and a total rainfall reduction of 52% (calculated from step 1 of the cascade model). The ability of this scenario to represent accurately LGM rainfall conditions depends on the quality of the sea-surface temperature reconstruction used as forcing data, as well as on the climate model parameterization. The latter was optimized to reproduce South American climate under present conditions sufficiently well[31]. Furthermore, the total rainfall reduction is in agreement with recent observation-based paleoprecipitation reconstructions[30]. Supplementary Table 3 summarizes the different model settings, scenarios and initial conditions used to produce the figures.

**Data availability.** The data can be downloaded from http://www.un-spider.org/links-and-resources/data-sources/land-cover-map-glc2000-jrc (GLC2000 land-cover map), https://data.iac.ethz.ch/landflux/ (LandFlux-EVAL evapotranspiration), http://hydrology.princeton.edu/data.pdsi.php (Potential evapotranspiration), ftp://ftp.glcf.umd.edu/glcf/Global_VCF/Collection_5/ (MOD44B v5 TC), http://www.csr.ufmg.br/simamazonia/ (Amazonian vegetation classification for 2003)

http://mirador.gsfc.nasa.gov/cgi-bin/mirador/presentNavigation.pl?tree=project data set = TRMM_3B42_daily.007 project = TRMM dataGroup = Gridded version = 007 (TRMM 3B-42 v7 precipitation). The precipitation data from CRU, GPCC, GPCP and CPC can be downloaded using links given in a previous study (ref. 75) and the merged synthesis product has been obtained from the author of that study (B.M.) upon request. The natural-vegetation-cover classification is accessible from Instituto Nacional de Pesquisas Espaciais (INPE)—Centro de Previso de Tempo e Estudos Climticos (CPTEC). The moisture recycling networks and the computer code of the models developed in this study are available from the corresponding author D.C.Z., upon reasonable request. A basic version of the WAM-2layers model is available through the link https://github.com/ruudvdent/WAM2layersPython.

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

## Acknowledgements

We thank Ruud van der Ent for developing and sharing the atmospheric-moisture tracking model used in this study, Egbert van Nes for providing code for Supplementary Fig. 1 and Brigitte Mueller for providing the precipitation multi-data set used in this study. We also thank Hermann Behling, A.J. Han Dolman, Jonathan F. Donges, Dieter Gerten, Milena Holmgren, Patrick Keys, Peter Puetz, Thomas Kneib, Marten Scheffer, Kirsten Thonicke and Liubov Tupikina for discussion and comments. We are thankful to Pierre Manceaux for his contribution to the design of the figures and to Alison Schlums for proof-read. This paper was developed within the scope of the IRTG 1740/TRP 2011/50151-0, funded by the DFG/FAPESP. D.C.Z. acknowledges the financial support from EU-FP7 ROBIN project under grant agreement 283093. H.M.J.B. acknowledges the financial support from FAPESP through grants number 11/50151-0 and 13/50510-5. L.W.-E. acknowledges the financial support from The Swedish Research Council Formas through grant number 1364115. A.S. acknowledges the financial support from SENSE Research School, A.R. the EU-FP7 project AMAZALERT (ProjectID 282664) and M.H. the project Microsoft/FAPESP 2013/50169-1, and C.F.S. the financial support by the German Federal Ministry for the Environment, Nature Conservation and Nuclear Safety (16_II_148_Global_A_IMPACT).

## Author contributions

D.C.Z. and C.-F.S. conceived the study, D.C.Z. conducted the analysis and wrote the paper with the support of all co-authors. A.R. supervised the study.

## Additional information

**Competing financial interests:** The authors declare no competing financial interests.

**Publisher's note**: 

