## [Peer Review File · Nature Communications]

Reviewers' Comments:

Reviewer #1 (Remarks to the Author)

On the physical side, the evaporation model appears over-simplified. The authors include Dai (Nature Climate Change) as a reference but fail to include Sheffield et al. (Nature) let along the Trenberth et al. (Nature Climate Change) commentary of both: when taken together these show how insights about evaporation are sensitive to the choice of model, parameters and data.

On another note, the authors make a statement that "although this approach is theoretically suited to anticipate critical transitions in complex systems despite the lack of understanding of the underlying mechanisms, [reference to Scheffer et al. 2012 in Science] it has never been applied to the Amazon forest so far". The 2012 review paper in Science is not really a proof or a validation of the ability of complex networks to anticipate critical transitions in this case especially given the liberal assumptions made for dynamical processes which are extremely sensitive to these assumptions.

On the complex network side, there is almost no discussion of the literature [other than the Watt 2002 paper in PNAS]. The 2002 PNAS paper is rather far removed from this domain however. In this paper, basic considerations like the ability to capture time lag effects are not discussed. The concept of resilience (defined as the "probability to find forest for a given rainfall regime?") appears ill-defined and ad hoc (both in the main and in the Methods sections).

One question that needs to be answered clearly is to what extent the results and insights may be artifacts of the imposed network structure and to what extent they represent reality. What confidence do we have in their ability to represent reality? Is the "importance of spatial heterogeneity for the stability" a property of complex networks or ecosystems or both? What about the insensitivity to the choice of thresholds? When determining that "heterogeneity in forest resilience thresholds weakens the tendency of high-order cascades" how are spatial or temporal correlations considered if at all?

Given the issues above, the confidence in the results in insights (e.g., based on link removal experiments) cannot be too high. Questions could even be raised on whether the so called sensitivity analysis that the authors claim to perform with an overly simplified model can yield verifiable insights.

The ability to bring together advanced empirical tools such as complex networks with system dynamics and process understanding is useful, and perhaps critical to understand the behavior of complex processes in geophysics, including perhaps tipping points in Amazonia. However, this manuscript falls short on the adaptation of the empirical tool as well as the application of the process knowledge, and appears innocent of the nuances and the sensitivities of the underlying assumptions. In general, such approaches may need a deeper consideration of falsifiability to ensure that the results and insights are not artifacts of the imposed (in this case complex network) structure but a fundamental discovery about (in this case ecosystem) science.

Reviewer #2 (Remarks to the Author)

Overall, I think this is a very interesting study which is very original in approach. The modelling approach provides a simple, yet elegant framework for evaluating the resilience of Amazonian forests to the combined threats of climate change and land use change. The manuscript (including the main text and supplementary information) is well-written and well-presented. The main results are that vegetation-atmosphere feedbacks amplify the risk of forest loss under future drought scenarios and that ecosystem heterogeneity makes the Amazon more resilient to future

precipitation reductions.

General Comments

1. My main comment is that, despite the originality of the approach, the authors need to demonstrate more clearly the new insights that this approach allows for. None of the key results presented in the abstract are novel per se, as they have previously been reported by studies using coupled climate model studies or dynamic global vegetation modelling studies. There is a long history of studies that have looked at the effects of deforestation on Amazonian rainfall. The role of ecosystem heterogeneity in conferring stability to the Amazon forest system has also been documented clearly recently. The advantages of the complex network approach relative to more traditional approaches and the new questions it can help to answer need to be much clearer.

2. Overall, I found the methodology very interesting and well-described. However, I found it less clear to understand exactly how the vegetation transition process occurs in the model, especially with regards to how 'equilibrium' resilience thresholds based on observed tree cover distributions are translated into vegetation type transitions in future precipitation scenarios. I understand that 'forest resilience' values were obtained by fitting a logistic regression model to data on the frequency distribution of different vegetated states (eq. 16-18 in SI) and that different threshold resilience values were chosen for determining forest-savanna shifts. However, it was not clear to me what the timescales involved in eliciting shifts were. Was it just assumed that a shift occurred following one year of decreased resilience below the threshold or were time lags built into the model in some way? Assumptions of the amount of time a forest patch can persist below resilience thresholds before shifting to an alternate state could have a big impact on the model results. The authors should clearly present how they dealt with the issue of timescales of vegetation shifts and provide justification for this. I would also recommend additional analysis to explore the sensitivity of the results to these assumptions.

Specific comments

Abstract:

Lines 5-6: The sentence 'there is growing evidence that forests might tip towards savanna with reduced rainfall' seems a bit too strong and should be toned down. For example, most DGVM studies suggest a general pattern of resilience of tropical forests to future climate change, largely due to CO₂ fertilization (e.g. Huntingford et al. 2013, Rammig et al. 2010).

Line 12: Should this be 'deforestation' instead of 'degradation'? For many people, degradation is more related to logging practices rather than clearfelling of forests.

Introduction:

Lines 14-15: This conveys the false impression that 'die-back' results in a large number of future vegetation simulation models. In fact, it is only simulated in a small number of model simulations, usually under a HadCM3 climate.

Lines 17-22: You state that the resilience of vegetation is affected by 'mean' precipitation, but surely other aspects of the precipitation regime (seasonality, frequency of extreme events) are as important if not more? Also, I would not use the expression of forest 'degrading' to another state - this controversially suggests that non-forest biomes have limited value. I suggest using 'shift' instead of 'degrade'.

Results:

Line 49: I am not sure that I would agree that your results are insensitive to the choice of

resilience thresholds. Extended Figure 2D suggests that the fraction of remaining forest and the share of the cascading effect are quite sensitive to the assumed value of the 'resilience threshold'.

Methods:

Generally well-described. Modelling framework is clearly presented.

Discussion:

Lines 85-87: The authors mention CO₂ and temperature as important variables their model does not consider. However, fire, perhaps one of the most critical factors in driving vegetation transitions, also needs to be mentioned. Is it the view of the authors that the empirical formulation of forest resilience based on precipitation data implicitly accounts for fire effects? If so, this needs to be discussed.

Supplementary Information

The SI requires one more careful read through. There are a number of small typos that need correction (e.g. lines 135, 195, 312).

Reviewer #3 (Remarks to the Author)

The authors study the interactions and feedbacks between forest cover and rainfall over the Amazon. Specifically the study explores the potential for a self amplifying rainfall-forest feedback where reduced rainfall impacts forests and reduced forest cover impacts rainfall. Most previous studies have been restricted to studying one part of the system - either the impacts of forests on rain, or the impacts of rain on forests. Through attempting to study both parts of this feedback the study makes some important progress in understanding the whole system. This is therefore an important study. However, description of the methods and presentation of the key results is confusing and this makes the manuscript difficult to follow in a number of places. Reliance on figures in the Extended Data to support crucial parts of the story makes the paper tricky to understand. The authors should give careful thought to how to present their study more clearly. I have given some examples of where this could be improved below.

Line 27-28 I could not see where the reconstruction of moisture recycling or atmospheric moisture tracking was explained in the Methods. This is a crucial component of the analysis and needs careful explanation.

Line 41-45. I found this section confusing and difficult to follow. A more logical presentation of the figures might help: Figure g is explained before Figure 3a-f. One sentence here relies on results in Figures 3, 10 and 4 in Extended Data. Is it possible to order figures in Extended Data more logically, or present some of important results into the main paper?

Line 42. Is the reduction really exponential?

Line 56-57. Should this be Fig. 3d?

Line 73. This section is confusing. Again there is heavy reliance on Extended Data and Supplementary Methods to understand the main points of this figure. To understand Fig. 4a the reader is directed to the Methods and Extended Data Fig. 6 and Extended Data Fig. 2 and the Supplementary Methods. This is an important part of your analysis - is it possible to explain more fully in the main text and Fig. Caption. I did not fully understand Fig. 4b and what is shown by the dashed and solid lines. Please carefully clarify what is shown here. I may have misunderstood, but the 50% increase in forest loss does not seem to match black line in Fig. 4b (varies from 0 to 16%).

Line Change "Scondly"

Line 106-154. General comments on methods. What is the spatial resolution of the analysis (in Fig. 2 and 3)? What determines this resolution? How do changes in moisture input (from ET) change calculated rainfall? This is not well explained in the Methods, but is crucial to your analysis.

Line 126. Please explain how Fig. 3a shows the model reproduces forest cover under historical conditions.

Line 140. What is variable C?

Figure 3: g: The change in oceanic moisture flux should be negative. Caption "e, f", Spelling of "Break-down"

Extended data, Figure 7 b. Treeless should be "TC<5%"

Extended data, Figure 10a. Colour bar: Is this figure in percent? If so, percentages in (c) don't seem to match panels.

Reviewer #4 (Remarks to the Author)

A. Summary of the key results: This paper uses a novel cascading complex network approach to examine the projected sensitivity of the Amazon rainforest to future climate change.

B. Originality and interest: The approach appears novel, and the potential for the Amazon to be a "tipping element" as climate changes implies wide interest.

C. Data and methodology: There are significant issues with the the data and methodology. The most significant problem is that "history" appears to be defined over a 6 year period, which is insufficient to provide any context to what the authors are trying to convey. This, coupled with a relentless focus on future projections rather than on past behavior fatally compromises the paper. Specific concerns:

1) Historical precipitation data over the past 100+ years suggest remarkable stability for the entire basin (75 °W - 50 °W, 15°S-5°N). Neither the CRU TS3.23 nor the GPCC V7 precipitation show significant long-time scale variability beyond some multi-decadal oscillations. Insofar as there has been extensive deforestation in the region, this begs the very necessary question of why there isn't a stronger signal already emerging.

2) Pollen studies (e.g., Haberle (1997) in Proceedings of the Ocean Drilling Program, Scientific Results vol. 155) seem to suggest that the Amazon became drier with the expansion of the savannah during Quaternary glacial periods. The authors propose a similar expansion in response to warming in this work. If this is the situation, then they are trying to model a true second-order impact of climate change, with the system currently in the bottom of what effectively amounts to being a potential well, with the authors more-or-less trying to model the curvature of that well. No climate model currently in existence can do this - they have enough difficulties with capturing leading order changes (e.g., warming rates). This begs the question of whether the authors have tried their approach on paleoclimate data to see if they reproduce anything similar to the observed change in Amazonian vegetation (e.g., using model simulations from the PMIP program)? This would give credence to the validity of the author's approach.

D. Appropriate uses of statistics and uncertainties: The internal statistics appear to be fine, but as noted above the authors are trying to characterize something that is very poorly constrained.

E. Conclusions: Given the above along with the relentless focus on projections, there are

significant concerns with robustness.

F. Suggested improvements: Model the past century, perhaps using 20C reanalysis data (available from ECMWF or NOAA), and look at glacial behavior using PMIP simulations

G. References: Fine

H. Clarity: Well written, but flawed as noted above.

Reviewer #5 (Remarks to the Author)

General Comment

The manuscript aims at studying the risk of self-amplified Amazon forest loss under a drastic intensification of the dry season using a novel network approach. This is a pretty relevant topic addressing the role of internal dynamics (climate-vegetation feedbacks) in Amazonia. The results are certainly worth publishing after a major revision, and provided that the authors satisfactorily address and take on board the following comments.

Specific Comments:

1. Abstract. Line 8. "Substantial risk"? Please provide estimates and uncertainty measures.
2. Abstract. Line 10. "This risk is reduced by increased spatial variability in forest's sensitivity to altered rainfall regimes." Reduced? How much? Altered rainfall regimes? Please provide estimates and uncertainty measures.
3. Lines 16-17. "The interactions between vegetation and regional climate that could lead to such dynamic are depicted in Fig. 1." This statement is related to an initial forest loss triggered by decreasing oceanic moisture inflow. What might be the cause of such a decrease in oceanic moisture inflow?
4. Line 18. Figure 2 is mentioned at the end of the line. This section serves as an introduction of the manuscript, but it is already showing results (Fig 2). Is it a previous result? If so, please provide reference. If not, the authors must explain what data and methods were used to produce this map. At any rate, please explain what are you trying to conveying with this figure at this point in the manuscript.
5. Lines 108-109. "We initialize 1000 ensemble simulations in which each grid cell has a resilience threshold randomly sampled from a normal distribution with mean Φ and standard deviation σ The thresholds are fixed for the duration of the simulation." The authors might want to discuss their results from the viewpoint of multiple testing.
6. Lines 112-113. "Moisture from oceanic and continental origin propagates through the network on a monthly time scale." What is the rationale behind this assumption having in mind that the timescale of precipitation recycling ranges from 5 to 12 days in Amazonia (van der Ent & Savenije, Atmos. Chem. Phys., 11, 1853-1863, 2011)? This assumption should be supported by an in-depth analysis and discussion of the linkages between the space and time scales of hydrological, climatic and vegetation processes in Amazonia. I think this is the main limitation of the manuscript, deserving a major revision.
7. Lines 131-132. "Quantifying cascading effects. We compare a fully coupled vegetation-atmosphere system ("cascade-mode on") with a one-way coupled system in which changes in vegetation states do not alter evapotranspiration rates ("cascade-mode off")." Does this procedure assume that precipitation remains the same regardless of the kind of vegetation? Please justify it.

8. Figure 3 a and b show results for a long-term mean annual rainfall regime estimated with 7 years worth of data (1989-1995). Such a short period of time hardly constitutes a valid time span for a long-term mean estimate.

9. Figure 3. Caption. "... as a function of monthly oceanic moisture inflow reduction during the extended dry season (ΔP_{ocean})." This statement and approach base the study on an oceanic moisture inflow reduction, without any specific cause. It somehow contradicts the aim of the manuscript, which is focused on the study of the role of internal dynamics (vegetation-atmosphere feedbacks), and not on external forcings.

10. I strongly suggest that the authors include a Discussion of Results section. The manuscript is way too comprised to fully understand what the authors do and conclude from the methods applied.

11. The Extended Data Figures comprise a lot of information. Some of them are not mentioned or (properly) discussed in the manuscript. Also, it seems that some of them are pretty relevant for the manuscript, and should not be left out as a supplementary material.

Reviewer #1 (Remarks to the Author):

1. On the physical side, the evaporation model appears over-simplified. The authors include Dai (Nature Climate Change) as a reference but fail to include Sheffield et al. (Nature) let along the Trenberth et al. (Nature Climate Change) commentary of both: when taken together these show how insights about evaporation are sensitive to the choice of model, parameters and data.

We thank the reviewer for pointing out the importance of assessing the sensitivity of our evapotranspiration estimations to the choice of model, parameters and input data. We performed additional analyses using independent evapotranspiration datasets from different categories (see "Input data" in Method section) and using a different evapotranspiration model (generalized linear model, see Supplementary Information Sect. 1.2.3). The choice of input data and model does not affect our main findings as shown in the Supplementary Figures and Table (Table S.1, Figs. S3, S4, S5) and discussed in the Supplementary Information (SI Sect. 2). This is also mentioned in the main text (L. 57-58, L. 65-66, L. 71-72, L. 145-146). We also account for uncertainties in parameter estimates by showing results obtained using the 95% confidence intervals (e.g., Fig. 4).

In the main text, we replaced the reference to Dai et al. (Nature, 2012) by the more recent and updated study from Trenberth et al. (Nature Climate Change) (see ref. 53, L. 129) and include also the reference to Sheffield et al. (Nature) (see ref. 52, L. 146).

2. On another note, the authors make a statement that "although this approach is theoretically suited to anticipate critical transitions in complex systems despite the lack of understanding of the underlying mechanisms, [reference to Scheffer et al. 2012 in Science] it has never been applied to the Amazon forest so far". The 2012 review paper in Science is not really a proof or a validation of the ability of complex networks to anticipate critical transitions in this case especially given the liberal assumptions made for dynamical processes which are extremely sensitive to these assumptions.

We agree with the reviewer that the ability of complex networks to anticipate critical transitions as stated by Scheffer et al., 2012 is not guaranteed *a priori* for the Amazon forest. In the revised manuscript, we removed this sentence.

3. On the complex network side, there is almost no discussion of the literature [other than the Watt 2002 paper in PNAS]. The 2002 PNAS paper is rather far removed from this domain however.

According to the reviewer's suggestion, we added more information on complex networks and their applications in the text (L. 35-37).

In our submitted manuscript, we cited only Watts et al. (2002) (ref. 33), because his simple model to study cascade dynamics in complex networks (called "Watts model") has largely inspired our methodology. Watts et al. (2002) modelled cascades that are found in social systems such as spread of innovations. Typically, individuals must decide between two alternative actions and their decisions depends explicitly on the actions of other individuals with whom they interact. Watts' model is based on a random network, in which each node can shift between alternative states depending on the states of their neighbors according to a simple threshold rule. More specifically, a random threshold is assigned to each node. A node shifts from state 0 to state 1 if the fraction of its neighbors that are in state 1 exceeds the specified threshold. Cascade dynamics are analyzed by setting initial population in state 0 and perturbing a small fraction of nodes that are switched to state 1. The population evolves at successive time steps with nodes updating their status according to the above thresholds, which are fixed for the duration of the simulations. The average time required for a cascade to reach steady-state, as well as the average number of nodes that have shifted, measured for 1,000 realizations of the dynamics, indicate cascades' sizes. The effect of spatial heterogeneity is analyzed by varying standard deviation in thresholds' distribution. We adopt a very similar approach, except that 1) our networks are not random and 2) our threshold rule is not related to the fraction of the neighbors that are in a given state, but rather to local forest resilience that depends on the total incoming moisture. We explain better how our model relates to Watts model in the revised manuscript (L. 193-198).

4. In this paper, basic considerations like the ability to capture time lag effects are not discussed.

We thank the reviewer for raising this important comment and refer to our response to comments from reviewer 2 (see comment 2).

5. The concept of resilience (defined as the "probability to find forest for a given rainfall regime"?) appears ill-defined and ad hoc (both in the main and in the Methods sections).

We thank the reviewer for pointing out the need to better explain the concept of forest resilience in our manuscript. In the manuscript (L.20), we have defined resilience as the ability of the vegetation to recover from perturbations (with reference to Verbesselt et al. 2016, ref. 12). The formulation used later in the text was confusing and we have rephrased the text (L. 51-52).

6. One question that needs to be answered clearly is to what extent the results and insights may be artifacts of the imposed network structure and to what extent they represent reality. What confidence do we have in their ability to represent reality?

We thank the reviewer for raising the important question of our confidence in the ability of our moisture recycling networks to represent real processes. This question has been addressed in detail in a previous study analyzing the network's topology (Zemp et al., 2014, ref. 23). For example, the sum of all incoming arrows pointing towards the Amazon region quantifies the well-known regional recycling ratios (Eltahir and Bras, 1994, ref. 21) and compared well with previous estimates based on different methodologies (see Table 2 in Zemp et al., 2014). Similarly, the sum of all outgoing and incoming arrows in the entire networks (called "out-strength" and "in-strength" in complex network terminology, respectively) (Figs. 1 and 2 in Zemp et al., 2014) represent continental recycling ratios and agreed with previous estimates (van der Ent et al., 2010, ref. 22). Finally, spatio-temporal variability of the direct source of rainfall over the La Plata basin inferred from moisture recycling networks (Zemp et al., 2014) were in line with previous estimates based on Lagrangian atmospheric moisture tracking algorithm relying on 10-days wind back-trajectories (Drumond et al. 2008 and 2014). Furthermore, we would like to mention that moisture recycling networks constructed based on different time periods (1989-1995 and 2000-2010) had similar topologies (Zemp et al. 2014), suggesting that the networks are not very sensitive to the input data and time period considered. In our revised manuscript, we add one sentence following the description of moisture recycling networks: "The ability of such networks to represent real moisture recycling processes has been shown in a previous detailed analysis of networks' topology (Zemp et al., 2014)" (L.41-42).

Drumond, A., Nieto, R., Gimeno, L. and Ambrizzi, T. (2008). A Lagrangian identification of major sources of moisture over Central Brazil and La Plata Basin. *J. Geophys. Res.*, 113, D14128, doi:10.1029/2007JD009547.

Drumond, A., Marengo, J., Ambrizzi, T., Nieto, R., Moreira, L., and Gimeno, L. (2014). The role of the Amazon Basin moisture in the atmospheric branch of the hydrological cycle: a Lagrangian analysis, *Hydrol. Earth Syst. Sci.*, 18, 2577-2598, doi:10.5194/hess-18-2577-2014.

7. Is the "importance of spatial heterogeneity for the stability" a property of complex networks or ecosystems or both?

We thank the reviewer for this comment. Given the above results, we are confident that properties emerging from moisture recycling networks reveal properties of the Amazon vegetation-rainfall system. However, we understand the concern of the reviewer and tuned down our claims in the revised manuscript (using the expressions "our results suggest" L.9 and "seems to" L. 106).

8. What about the insensitivity to the choice of thresholds?

Our results regarding the effect of heterogeneity are insensitive to the choice of thresholds, as shown in Fig. S9 (mentioned L. 104).

9. When determining that "heterogeneity in forest resilience thresholds weakens the tendency of high-order cascades" how are spatial or temporal correlations considered if at all?

This is an interesting question. The effect of spatial correlation could be partly answered by investigating how the spatial resolution of the networks affect our results. However, this resolution is determined by the settings of the atmospheric moisture tracking model WAM-2layers, which can not be easily changed. The effect of temporal correlation is not explicitly accounted for, since resilience thresholds are fixed for the duration of the simulation. In the revised manuscript, we mention this clearly in the Methods section (L. 211) and tuned down our claim as mentioned above (see point 7).

10. Given the issues above, the confidence in the results in insights (e.g., based on link removal experiments) cannot be too high. Questions could even be raised on whether the so called sensitivity analysis that the authors claim to perform with an overly simplified model can yield verifiable insights.

To address the reviewer's critique, we removed the link removal experiments in the revised manuscript. Instead, we have added a more detailed sensitivity analysis of our results regarding the effect of heterogeneity as stated above (see point 8).

11. The ability to bring together advanced empirical tools such as complex networks with system dynamics and process understanding is useful, and perhaps critical to understand the behavior of complex processes in geophysics, including perhaps tipping points in Amazonia. However, this manuscript falls short on the adaptation of the empirical tool as well as the application of the process knowledge, and appears innocent of the nuances and the sensitivities of the underlying assumptions. In general, such approaches may need a deeper consideration of falsifiability to ensure that the results and insights are not artifacts of the imposed (in this case complex network) structure but a fundamental discovery about (in this case ecosystem) science.

We understand the concern of the reviewer, as our results emerging from our experiments of increasing heterogeneity and links removal can not be "ground-based" verifiable. This is an obvious limitation of these kinds of experiments, which are commonly performed in the complex network approach. However, it can also be seen as an advantage, since it allows us to easily analyze the structural stability of networks. As stated above, we have analyzed the sensitivity of our results related to the effect of heterogeneity to model settings (Fig. S9) and removed the part on link removal experiments. We are convinced that the updated manuscript has a much more compelling argument.

Reviewer #2 (Remarks to the Author)

Overall, I think this is a very interesting study which is very original in approach. The modelling approach provides a simple, yet elegant framework for evaluating the resilience of Amazonian forests to the combined threats of climate change and land use change. The manuscript (including the main text and supplementary information) is well-written and well-presented. The main results are that vegetation-atmosphere feedbacks amplify the risk of forest loss under future drought scenarios and that ecosystem heterogeneity makes the Amazon more resilient to future precipitation reductions.

We thank the reviewer for the positive comments on the manuscript.

General Comments

1. My main comment is that, despite the originality of the approach, the authors need to demonstrate more clearly the new insights that this approach allows for. None of the key results presented in the abstract are novel per se, as they have previously been reported by studies using coupled climate model studies or dynamic global vegetation modelling studies. There is a long history of studies that have looked at the effects of deforestation on Amazonian rainfall. The role of ecosystem heterogeneity in conferring stability to the Amazon forest system has also been documented clearly recently. The advantages of the complex network approach relative to more traditional approaches and the new questions it can help to answer need to be much clearer.

We thank the reviewer for suggesting to strengthen the novelty of our findings. We have made the advantage of our approach clearer throughout the text (L. 109-111, L.139-145, L. 146-149).

2. Overall, I found the methodology very interesting and well-described. However, I found it less clear to understand exactly how the vegetation transition process occurs in the model, especially with regards to how 'equilibrium' resilience thresholds based on observed tree cover distributions are translated into vegetation type transitions in future precipitation scenarios. I understand that 'forest resilience' values were obtained by fitting a logistic regression model to data on the frequency distribution of different vegetated states (eq. 16-18 in SI) and that different threshold resilience values were chosen for determining forest-savanna shifts. However, it was not clear to me what the timescales involved in eliciting shifts were. Was it just assumed that a shift occurred following one year of decreased resilience below the threshold or were time lags built into the model in some way? Assumptions of the amount of

time a forest patch can pertain below resilience thresholds before shifting to an alternate state could have a big impact on the model results. The authors should clearly present how they dealt with the issue of timescales of vegetation shifts and provide justification for this. I would also recommend additional analysis to explore the sensitivity of the results to these assumptions.

We thank the reviewer for pointing out the need to better explain how vegetation shifts occur in our model in relation to timescales. In the revised manuscript, we have mentioned in the Methods section that forest nodes shift without any time lag (L. 209). We have also added a sentence: "As it does not resolve underlying processes of forest dieback, our method is not suited to provide information on the "real-world" time scale of self-amplified forest loss." (L. 111-112).

We appreciate the suggestion of the reviewer. However, introducing time lags for forest shifts would have no effect on our results. Indeed, nothing changes in our model from one year to another if there is no vegetation shift. This is because inter-annual variability in forcing conditions (deforestation and/or dry-season intensification) is not represented. In other words, each experiment of oceanic moisture inflow reduction (e.g., "end of 21st century" scenario) represents long-term mean conditions. While it would be very interesting to investigate the combined effects of rainfall inter-annual variability and time lags in forest shifts on the stability of the Amazon vegetation-rainfall system, it is beyond the scope of our study and will be addressed in another paper. We have mention this in the revised manuscript (l. 137-138).

Specific comments

Abstract:

Lines 5-6: The sentence 'there is growing evidence that forests might tip towards savanna with reduced rainfall' seems a bit too strong and should be toned down. For example, most DGVM studies suggest a general pattern of resilience of tropical forests to future climate change, largely due to CO₂ fertilization (e.g. Huntingford et al. 2013, Rammig et al. 2010).

We thank the reviewer for this remark. We have removed the expression "there is growing evidence that".

Line 12: Should this be 'deforestation' instead of 'degradation'? For many people, degradation is more related to logging practices rather than clearfelling of forests.

We agree with the suggestion of the reviewer and changed the word "degradation" to "deforestation" throughout the manuscript (e.g., L. 17).

Introduction:

Lines 14-15: This conveys the false impression that 'die-back' results in a large number of future vegetation simulation models. In fact, it is only simulated in a small number of model simulations, usually under a HadCM3 climate.

We thank the reviewer for this comment and changed the text accordingly: "Large-scale vegetation shifts resulting from reduced rainfall [...] might occur under 21st century climate change in combination with increasing deforestation, logging and fire." (L. 15-17)

Lines 17-22: You state that the resilience of vegetation is affected by 'mean' precipitation, but surely other aspects of the precipitation regime (seasonality, frequency of extreme events) are as important if not more?

In our calculation of the forest resilience, we included not only mean precipitation but also maximum climatological water deficit, as measure of the intensity of the dry season. We have changed the text in L.48 accordingly. The variability of forest resilience with changing MAP and MCWD is now also shown in Fig. 2. The potential effect of extreme drought events is briefly discussed (L. 128-132 and L. 137-138). Regarding the importance of mean precipitation for forest resilience mentioned in the introduction (L. 20), we refer to Poorter et al. (2016) (ref. 13) and Verbesselt et al. (2016) (ref. 12).

Also, I would not use the expression of forest 'degrading' to another state - this controversially suggests that non-forest biomes have limited value. I suggest using 'shift' instead of 'degrade'.

We agree with the reviewer and have changed the expression according to his/her suggestion.

Results:

Line 49: I am not sure that I would agree that your results are insensitive to the choice of resilience thresholds. Extended Figure 2D suggests that the fraction of remaining forest and the share of the cascading effect are quite sensitive to the assumed value of the 'resilience threshold'.

We agree that the sensitivity of our results to the choice of the resilience thresholds needs further explanation. As shown in the supplementary information (Figs. S6, S7, S8 and S9), the dynamic of self-amplified forest loss is robust for different choices of resilience thresholds. This is now mentioned in the text (L. 72, L. 104). We also explain that "While initial forest loss induced by reduced oceanic moisture inflow is sensitive to the underlying resilience thresholds, the additional forest loss attributed to vegetation-atmosphere feedbacks is more robust" (L. 73-74). In addition, we now systematically provide estimations for different realistic threshold values (for example in the abstract L. 12 and in the text L. 75 and L. 78).

Methods:

Generally well-described. Modelling framework is clearly presented.

We thank the reviewer for this positive comment.

Discussion:

Lines 85-87: The authors mention CO2 and temperature as important variables their model does not consider. However, fire, perhaps one of the most critical factors in driving vegetation transitions, also needs to be mentioned. Is it the view of the authors that the empirical formulation of forest resilience based on precipitation data implicitly accounts for fire effects? If so, this needs to be discussed.

We thank the reviewer for this comment. Indeed, the effect of fire is implicitly considered in our approach. We have made this more clear in the text (L. 54). The role of fire is also mentioned in the introduction (L. 17).

Supplementary Information

The SI requires one more careful read through. There are a number of small typos that need correction (e.g. lines 135, 195, 312).

Done.

Reviewer #3 (Remarks to the Author):

The authors study the interactions and feedbacks between forest cover and rainfall over the Amazon. Specifically the study explores the potential for a self-amplifying rainfall-forest feedback where reduced rainfall impacts forests and reduced forest cover impacts rainfall. Most previous studies have been restricted to studying one part of the system - either the impacts of forests on rain, or the impacts of rain on forests. Through attempting to study both parts of this feedback the study makes some important progress in understanding the whole system. This is therefore an important study. However, description of the methods and presentation of the key results is confusing and this makes the manuscript difficult to follow in a number of places. Reliance on figures in the Extended Data to support crucial parts of the story makes the paper tricky to understand. The authors should give careful thought to how to present their study more clearly. I have given some examples of where this could be improved below.

We thank the reviewer for these encouraging comments. We appreciate his/her efforts and the positive feedback to make our manuscript easier to understand and are happy to present here a revised version.

Line 27-28 I could not see where the reconstruction of moisture recycling or atmospheric moisture tracking was explained in the Methods. This is a crucial component of the analysis and needs careful explanation.

We have moved the description of the construction of moisture recycling networks from the supplementary information to the main text (see Methods, L. 177-187).

Line 41-45. I found this section confusing and difficult to follow. A more logical presentation of the figures might help: Figure g is explained before Figure 3a-f. One sentence here relies on results in Figures 3, 10 and 4 in Extended Data. Is it possible to order figures in Extended Data more logically, or present some of important results into the main paper?

We agree with the reviewer's comment and we now present all of the most important results in the main text (Fig. 2, ref in L. 50-52; Fig. 4, ref in L. 69-71). Besides, figures left in Supplementary Information were reordered.

Line 42. Is the reduction really exponential?

We replaced the word "exponentially" with "non-linearly" (L. 68).

Line 56-57. Should this be Fig. 3d?

We thank the reviewer and have corrected the manuscript accordingly.

Line 73. This section is confusing. Again there is heavy reliance on Extended Data and Supplementary Methods to understand the main points of this figure. To understand Fig. 4a the reader is directed to the Methods and Extended Data Fig. 6 and Extended Data Fig. 2 and the Supplementary Methods. This is an important part of your analysis - is it possible to explain more fully in the main text and Fig. Caption. I did not fully understand Fig. 4b and what is shown by the dashed and solid lines. Please carefully clarify what is shown here. I may have misunderstood, but the 50% increase in forest loss does not seem to match black line in Fig. 4b (varies from 0 to 16%).

This section has been removed from the revised manuscript (see also our response to reviewer 1 comment 10).

Line Change "Scondly"

Done.

Line 106-154. General comments on methods. What is the spatial resolution of the analysis (in Fig. 2 and 3)? What determines this resolution?

The spatial resolution in the analysis depends on the processes represented. We have briefly mentioned the resolution in the Methods section (L. 165, L. 171-176, L. 179) and added more information in the Supplementary Methods (SI Sect. 1.1). Furthermore, we have added an overview table (see Table S2) summarizing the spatial resolution of all the figures.

How do changes in moisture input (from ET) change calculated rainfall? This is not well explained in the Methods, but is crucial to your analysis.

We explain how evapotranspiration affects rainfall in the Methods section (L. 203-205) and provide estimates of the effect of complete Amazon forest loss on rainfall in the Results section (L. 56-66).

Line 126. Please explain how Fig. 3a shows the model reproduces forest cover under historical conditions.

This is now illustrated in new figures (Figs. S12 and S13) and better explained in the Method section (L. 212-214).

Line 140. What is variable C?

We corrected this typo in the revised version and apologize for the confusion. It should read A (carry-over factor).

Figure 3: g: The change in oceanic moisture flux should be negative. Caption "e, f", Spelling of "Break-down"

Done. Corrected throughout the manuscript.

Extended data, Figure 7 b. Treeless should be "TC<5%"

Done. Corrected throughout the manuscript.

Extended data, Figure 10a. Colour bar: Is this figure in percent? If so, percentages in (c) don't seem to match panels.

Indeed, the caption of Extended Data Fig. 10 was not in percent. We corrected this mistake in the revised manuscript (Table S1 and Figs. 3b, 4c, S3, S4 and S5d) and apologize for the confusion.

Reviewer #4 (Remarks to the Author)

A. Summary of the key results: This paper uses a novel cascading complex network approach to examine the projected sensitivity of the Amazon rainforest to future climate change.

B. Originality and interest: The approach appears novel, and the potential for the Amazon to be a "tipping element" as climate changes implies wide interest.

C. Data and methodology: There are significant issues with the the data and methodology. The most significant problem is that "history" appears to be defined over a 6 year period, which is insufficient to provide any context to what the authors are trying to convey. This, coupled with a relentless focus on future projections rather than on past behavior fatally compromises the paper.

We thank the reviewer for pointing out the need to use a longer time series to constrain our model before performing simulation of environmental perturbations. For the construction of moisture recycling networks and the evapotranspiration model, we used in our submitted manuscript the "short" product from Landflux-EVAL (1989-1995) that had the advantage of being a merged product from 40 different evapotranspiration datasets (Mueller et al. 2012, ref. 60). However, we understand the concern of the reviewer and used in the revised manuscript the "long" product from Landflux-EVAL covering 17 years (1989-2005) and containing 14 different datasets (including ground-based and satellite observations, land-surface model outputs and reanalysis data) (see Methods section, L. 161-163). We think that this dataset provides the best balance between length of the time period and number of independent products from different categories included. The latter aspect is crucial, as different methods have different advantages and difficulties in accurately estimating evapotranspiration (see comment 1 from reviewer 1). The change of evaporation dataset has, however, not significantly affected our main results.

To estimate the parameters of the logistic regression model for our forest resilience calculation, we used in the submitted manuscript precipitation data from the Tropical Rainfall Measuring Mission (TRMM 3B42, Huffman et al, 2007, ref. 63) covering the period 2000-2012. This dataset was generated using radar and microwave instruments from satellites in combination with ground-based measurements at a resolution of 0.25° longitude and latitude, and is considered as the best high-resolution precipitation datasets (Kim et al. 2013, ref. 66). Therefore, we still use it in the revised manuscript to determine the variability of present-day Amazon forest resilience with changing rainfall regime (Figs. 2b, Fig. S1). However, to fully account for climate natural variability when simulating the effect of perturbations on vegetation distribution, we constrain the regression model using the Climate Research Unit's (CRU) high-resolution monthly data covering the period 1961-2012. This product is an interpolation of ground-based observation at 0.5° longitude and latitude grid (Mitchell et al. 2005). This is mentioned in the method section (L. 172-176) and the output of the regression using both datasets are shown in the Supplementary Methods (SI Table 1).

Mitchell, T. D. and Jones, P. D. (2005). An improved method of constructing a database of monthly climate observations and associated high-resolution grids. *Int. J. Climatol.*, 25: 693–712. doi:10.1002/joc.118.

Specific concerns:

1) Historical precipitation data over the past 100+ years suggest remarkable stability for the entire basin (75 °W - 50 °W, 15°S-5°N). Neither the CRU TS3.23 nor the GPCP V7 precipitation show significant long-time scale variability beyond some multi-decadal oscillations. Insofar as there has been extensive deforestation in the region, this begs the very necessary question of why there isn't a stronger signal already emerging.

We thank the reviewer for this comment. Ground-based and satellite observations have led to contrasting results regarding Amazonian rainfall change, and more sampling would be needed to conclude on the existence of a robust signal (Lawrence et al. 2015, ref. 40). Moreover, modelling studies suggest that rainfall reduction from past deforestation is less than natural variability, potentially explaining the absence of a strong signal already emerging (Spracklen et al. 2015, ref. 41). This is in agreement with our results, suggesting that the reduction of Amazonian rainfall induced by forest loss become strong if combined with oceanic moisture inflow reduction.

2) Pollen studies (e.g., Haberle (1997) in *Proceedings of the Ocean Drilling Program, Scientific Results* vol. 155) seem to suggest that the Amazon became drier with the expansion of the savannah during Quaternary glacial periods. The authors propose a similar expansion in response to warming in this work. If this is the situation, then they are trying to model a true second-order impact of climate change, with the system currently in the bottom of what effectively amounts to being a potential well, with the authors more-or-less trying to model the curvature of that well. No climate model currently in existence can do this - they have enough difficulties with capturing leading order changes (e.g., warming rates). This begs the question of whether the authors have tried their approach on paleoclimate data to see if they reproduce anything similar to the observed change in Amazonian vegetation (e.g., using model simulations from the PMIP program)? This would give credence to the validity of the author's approach.

We thank the reviewer for raising this interesting point. On the one hand, we have to deal here with a trade-off between the length of the time series and the quality of the data, as raised in our response above (see point C). On the other hand, this comment refers to a very interesting question related to the point that the Amazon basin experienced numerous drastic droughts on a millennial time scale, and in particular during the Last Glacial Maximum (LGM). Based on this suggestion, we generated an additional scenario ("LGM" scenario, see Methods L. 139-146) and show the resulting self-amplified forest loss (Fig. 5d,5e,5f, see also ref in L. 80-87). These new results validate our approach and broaden our storyline (see L. 118-126).

D. Appropriate uses of statistics and uncertainties: The internal statistics appear to be fine, but as noted above the authors are trying to characterize something that is very poorly constrained.

E. Conclusions: Given the above along with the relentless focus on projections, there are significant concerns with robustness.

F. Suggested improvements: Model the past century, perhaps using 20C reanalysis data (available from ECMWF or NOAA), and look at glacial behavior using PMIP simulations

We thank the reviewer for these suggestions. Precipitation from reanalysis datasets exhibit systematic biases and poorly resolve convectively coupled equatorial waves (Kim et al. 2013, ref. 66), which control a substantial fraction of tropical rainfall variability (Kiladis et al., 2009). Furthermore, imbalance of the moisture budget is found due to inconsistencies in precipitation and evapotranspiration estimates (Lorenz et al. 2012). To better constrain our model, we prefer to use CRU precipitation data covering a period over 52 years in the past for the calculation of forest resilience, as well as the "long" evapotranspiration product from LandFlux-EVAL covering 17 years for construction of moisture recycling networks and calibration of the evapotranspiration model. Consistencies of the datasets are shown in Budyko plots (Fig. S11).

Pollen-based climate reconstructions that were used for comparison with global model simulations in PIMP are missing for South America (Bartlein et al. 2011). Therefore, we propose to base our analysis on simulations from a regional climate model that has been parameterized to reproduce South American climate (Cook et al. 2006, ref. 29).

Bartlein, P. J., Harrison, S. P., Brewer, S., Connor, S., Davis, B. A. S., Gajewski, K., Guiot, J., Harrison-Prentice, T. I., Henderson, A., Peyron, O., Prentice, I. C., Scholze, M., Seppa, H., Shuman, B., Sugita, S., Thompson, R. S., Vial, A. E., Williams, J., and Wu, H. (2011), Pollen-based continental climate reconstructions at 6 and 21 ka: A global synthesis, *Clim. Dyn.*, 37, 775-802.

Lorenz, C. and Kunstmann, H. (2012). The hydrological cycle in three state-of-the-art reanalyses: Intercomparison and performance analysis. *J. Hydrometeor.*, 13, 1397–1420, doi: 10.1175/JHM-D-11-088.1.

Kiladis, G. N., M. C. Wheeler, P. T. Haertel, K. H. Straub, and P. E. Roundy (2009), Convectively coupled equatorial waves, *Rev. Geophys.*, 47, RG2003, doi:10.1029/2008RG000266.

G. References: Fine

H. Clarity: Well written, but flawed as noted above.

Reviewer #5 (Remarks to the Author):

General Comment

The manuscript aims at studying the risk of self-amplified Amazon forest loss under a drastic intensification of the dry season using a novel network approach. This is a pretty relevant topic addressing the role of internal dynamics (climate-vegetation feedbacks) in Amazonia. The results are certainly worth publishing after a major revision, and provided that the authors satisfactorily address and take on board the following comments.

Specific Comments:

1. Abstract. Line 8. "Substantial risk"? Please provide estimates and uncertainty measures.

We thank the reviewer for this suggestion. In the revised manuscript, we provide estimates and uncertainty measures in the abstract (L. 12) and the main text (L. 75, L. 78, L. 86, L. 91).

2. Abstract. Line 10. "This risk is reduced by increased spatial variability in forest's sensitivity to altered rainfall regimes." Reduced? How much? Altered rainfall regimes? Please provide estimates and uncertainty measures.

We appreciate the suggestion to be more transparent on our findings. However, the effect of heterogeneity is quantified based on cascade sizes, which is a different metric than the area forest loss due to self-amplified effects (see Methods L. 222-228). To keep the abstract short (up to 150 words permitted in Nature Communications), we do not provide quantitative estimates associated with the effect of heterogeneity, but rather focus on the implication of this result. We provide quantitative estimates to illustrate our statement in the main text (L. 103-104). We also explain that this experiment is performed under complete breakdown of dry-season oceanic moisture inflow (L. 102-103).

3. Lines 16-17. "The interactions between vegetation and regional climate that could lead to such dynamic are depicted in Fig. 1." This statement is related to an initial forest loss triggered by decreasing oceanic moisture inflow. What might be the cause of such a decrease in oceanic moisture inflow?

We thank the reviewer for reminding us to explain the reason why we are interested in dry-season ocean moisture inflow reduction. This is now mentioned in the introduction (L. 26-31).

4. Line 18. Figure 2 is mentioned at the end of the line. This section serves as an introduction of the manuscript, but it is already showing results (Fig 2). Is it a previous result? If so, please provide reference. If not, the authors must explain what data and methods were used to produce this map. At any rate, please explain what are you trying to conveying with this figure at this point in the manuscript.

We thank the reviewer for this important comment. In the revised manuscript, Fig. 3a is already mentioned in the introduction (L. 41) because it is similar to previous results (ref. 22), although here use different input datasets. We have now provided the appropriate reference in the figure caption and in the text. All other figures are new results and therefore are described in the Results section.

5. Lines 108-109. "We initialize 1000 ensemble simulations in which each grid cell has a resilience threshold randomly sampled from a normal distribution with mean Φ and standard deviation σ The

thresholds are fixed for the duration of the simulation." The authors might want to discuss their results from the viewpoint of multiple testing.

We apologize for the misleading formulation in our text. In our case, we do not have a multiple testing problem because all our 1000 ensemble members are truly unique (given that each grid cell is individually sampled). We have improved the description of threshold distribution in the Methods section of our revised manuscript (L. 198-199, L. 210).

6. Lines 112-113. "Moisture from oceanic and continental origin propagates through the network on a monthly time scale." What is the rationale behind this assumption having in mind that the timescale of precipitation recycling ranges from 5 to 12 days in Amazonia (van der Ent & Savenije, *Atmos. Chem. Phys.*, 11, 1853-1863, 2011)? This assumption should be supported by an in-depth analysis and discussion of the linkages between the space and time scales of hydrological, climatic and vegetation processes in Amazonia. I think this is the main limitation of the manuscript, deserving a major revision.

We thank the reviewer for raising this important question. We consider the relevant timescale of precipitation recycling in our approach, since the atmospheric moisture tracking model applied in our study (WAM-2layers) performs water balance at the smallest temporal resolution of the input data (ERA-Interim reanalysis), which corresponds to 3 hours. The output of WAM-layers is then averaged to monthly moisture transport between grid cells, which is used to construct our networks. This is now better explained in the Methods section (L. 180-181). The rationale behind this procedure is that evapotranspiration rates of varying vegetation states differ mainly at seasonal scale as shown by flux tower measurements. This is better explained in the text (L. 201-203). We also mention now that the variables chosen to determine forest resilience are "the best hydrological indicators to explain the variability of vegetation distribution in the Tropics (Mahli et al. 2009, ref. 66)" (L. 48-49). Finally, we make now clear that "As it does not resolve underlying processes of forest dieback, our method is not suited to provide information on the "real-world" time scale of self-amplified forest loss." (L. 111-112). As an outlook of our study, we suggest that "Further efforts are needed to assess the effect of inter-annual rainfall variability on the stability of the Amazon vegetation-rainfall system and potential time lags in the response of the coupled system." (L. 137-138).

7. Lines 131-132. "Quantifying cascading effects. We compare a fully coupled vegetation-atmosphere system ("cascade-mode on") with a one-way coupled system in which changes in vegetation states do not alter evapotranspiration rates ("cascade-mode off")." Does this procedure assume that precipitation remains the same regardless of the kind of vegetation? Please justify it.

We thank the reviewer for this comment. In the revised manuscript, we have better explain the different model settings (L. 219-221, see also Fig. S2) in order to improve the clarity of the procedure.

8. Figure 3 a and b show results for a long-term mean annual rainfall regime estimated with 7 years worth of data (1989-1995). Such a short period of time hardly constitutes a valid time span for a long-term mean estimate.

This criticism has been raised by other reviewers. We now use CRU data covering a period over 52 years in the past for the calculation of forest resilience, as well as the "long" product from Landflux-EVAL covering 17 years for construction of moisture recycling networks and calibration of the evapotranspiration model. For a detailed explanation, we invite the reviewer to read our response to reviewer 4, comment C.

9. Figure 3. Caption. "... as a function of monthly oceanic moisture inflow reduction during the extended dry season (ΔP_{ocean})." This statement and approach base the study on an oceanic moisture inflow reduction, without any specific cause. It somehow contradicts the aim of the manuscript, which is focused on the study of the role of internal dynamics (vegetation-atmosphere feedbacks), and not on external forcings.

We thank the reviewer for this comment. We have better justified our experiment (L. 26-31). We also mention that "a decrease in oceanic moisture inflow could trigger vegetation-atmosphere feedbacks and lead to self-amplified forest loss." (see L. 24-25).

10. I strongly suggest that the authors include a Discussion of Results section. The manuscript is way too comprised to fully understand what the authors do and conclude from the methods applied.

We thank the reviewer for this suggestion. We have followed the recommended format for Nature Communications papers: one Results section, followed by one Discussion section. We have rearranged

the text and moved several explanations from the Supplementary Information to the main text. We hope that, in this way, it is possible to follow our conclusions.

11. The Extended Data Figures comprise a lot of information. Some of them are not mentioned or (properly) discussed in the manuscript. Also, it seems that some of them are pretty relevant for the manuscript, and should not be left out as a supplementary material.

We thank the reviewer for this suggestion. We have deleted several figures from the Supplementary Information that were not discussed in the manuscript, and moved several key results to the main text (see new Figs. 2, 3b, 4b and 4c).

REVIEWERS' COMMENTS:

Reviewer #1 (Remarks to the Author):

The authors have addressed many of my comments. As a minor aside, I would have still liked to see a few citations to the prior literature where complex networks have been used in similar disciplines. That could have helped set the stage. The major issue remains: "...to what extent the results and insights may be artifacts of the imposed network structure and to what extent they represent reality. What confidence do we have in their ability to represent reality?" The authors point to Zemp et al. (2014) and have added one sentence in the manuscript, but I am not convinced that the Zemp paper adequately addresses the issue. I would suggest the authors make a better effort to address this issue in the context of this problem. Overall though, the manuscript does contribute to the existing literature, and based on my detailed assessment of all the reviewers' comments (including my own) and the authors' response, I would still recommend publication.

Reviewer #2 (Remarks to the Author):

Overall, the authors have done a very good job of responding to the concerns raised by the reviewers. The description of the methodology is now clearer and the authors are also now more up-front about the limitations of the approach used.

My only remaining comment is that the 'drying' of the Amazon is somewhat oversold and needs to be toned down somewhat. For example, lines 127-128 refer to recent [drying] trends, but actually the Amazon seems to be getting wetter (see Gloor et al. 2013, *Geophysical Research Letters* 40:1-5) and most GCMs do not simulate a drier future for the Amazon, although they all simulate a warmer future.

Reviewer #3 (Remarks to the Author):

The authors have made comprehensive and detailed revisions to their manuscript to account for the referee comments. My comments to the previous version of this manuscript have been addressed. The revised manuscript is much improved and I have no further comments.

This is a novel analysis that advances our understanding of vegetation-atmosphere feedbacks. In my opinion the revised manuscript is suitable for publication.

Reviewer #4 (Remarks to the Author):

The revised manuscript answers many of my concerns with the initial submission, and is much improved.

The one outstanding concern is the question of modeling second order effects in the system - if the authors are to be believed, then the pre-industrial state of the Amazon truly represents an "optimum," with both LGM and climate change scenarios representing a reduction in forest cover over the Amazon. As stated in my initial review, it is difficult enough to attempt to capture first order (linear) responses to climate change, let alone a second order effect as made apparent in the forest cover images shown in Figure 5. The modeling approach is interesting and novel, but I am unconvinced whether such second order effects can be meaningfully captured. How are we to interpret extreme warm climates of the past (e.g., the Eocene, or even the early Pliocene) - did all forests die back in this situation due to these cascading effects?

Summary: Interesting, but very speculative.

Reviewer #1 (Remarks to the Author):

The authors have addressed many of my comments. As a minor aside, I would have still liked to see a few citations to the prior literature where complex networks have been used in similar disciplines. That could have helped set the stage. The major issue remains: "...to what extent the results and insights may be artifacts of the imposed network structure and to what extent they represent reality. What confidence do we have in their ability to represent reality?" The authors point to Zemp et al. (2014) and have added one sentence in the manuscript, but I am not convinced that the Zemp paper adequately addresses the issue. I would suggest the authors make a better effort to address this issue in the context of this problem. Overall though, the manuscript does contribute to the existing literature, and based on my detailed assessment of all the reviewers' comments (including my own) and the authors' response, I would still recommend publication.

We thank the reviewer for the suggestions. We added three citations of previous studies in which the Earth System has been represented as a complex network (l. 53-55).

Reviewer #2 (Remarks to the Author):

Overall, the authors have done a very good job of responding to the concerns raised by the reviewers. The description of the methodology is now clearer and the authors are also now more up-front about the limitations of the approach used.

My only remaining comment is that the 'drying' of the Amazon is somewhat oversold and needs to be toned down somewhat. For example, lines 127-128 refer to recent [drying] trends, but actually the Amazon seems to be getting wetter (see Gloor et al. 2013, *Geophysical Research Letters* 40:1-5) and most GCMs do not simulate a drier future for the Amazon, although they all simulate a warmer future.

We thank the reviewer for this constructive comment with which we agree. We have given greater details on the current knowledge regarding climate variability for the past and the future in the Amazon region (l. 39-40) and tuned down our claims regarding the drying (l. 42-43).

Reviewer #3 (Remarks to the Author):

The authors have made comprehensive and detailed revisions to their manuscript to account for the referee comments. My comments to the previous version of this manuscript have been addressed. The revised manuscript is much improved and I have no further comments.

This is a novel analysis that advances our understanding of vegetation-atmosphere feedbacks. In my opinion the revised manuscript is suitable for publication.

We thank the reviewer for this positive comment and we are happy to present our final version of the manuscript.

Reviewer #4 (Remarks to the Author):

The revised manuscript answers many of my concerns with the initial submission, and is much improved.

The one outstanding concern is the question of modeling second order effects in the system - if the authors are to be believed, then the pre-industrial state of the Amazon truly represents an "optimum," with both LGM and climate change scenarios representing a reduction in forest cover over the Amazon. As stated in my initial review, it is difficult enough to attempt to capture first order (linear) responses to climate change, let alone a second order effect as made apparent in the forest cover images shown in Figure 5. The modeling approach is interesting and novel, but I am unconvinced whether such second order effects can be meaningfully captured. How are we to interpret extreme warm climates of the past (e.g., the Eocene, or even the early Pliocene) - did all forests die back in this situation due to these cascading effects?

Summary: Interesting, but very speculative.

We thank the reviewer for these comments. Indeed, so far the scientific community has no clear understanding of the role of non-linear vegetation-atmosphere interactions in shaping the Amazon forest cover. Analysis of pollen and isotopic composition in sediments and speleothems do not allow to answer this question as the external and internal (i.e., regional feedbacks) drivers of climatic change are intertwined. Progress is being made to quantify these feedbacks using coupled models, but large uncertainties remain using deterministic approaches as mentioned in our introduction (l. 48-49). In this respect, we are confident that our study, which is based on empirical evidence, allows to evaluate the role of regional vegetation-atmosphere interactions with the first and second order response including the cascading effect. As we said at the beginning of the discussion (l. 148-154), our study should be seen as a sensitivity analysis rather than a projection of the system dynamics, as it omits several key processes. In the revised manuscript (l. 155-158), we have changed the discussion of our results for the Last Glacial Maximum in order to better emphasize the limitation of our study.